

Assessment of satellite observation-based wildfire emissions inventories using
TROPOMI data and IFS-COMPO model simulations
Adrianus de Laat[1,#]
Vincent Huijnen[1]
Niels Andela[2]
Matthias Forkel[3]
[#] corresponding author, laatdej@knmi.nl
[1]Royal Netherlands Meteorological Institute, de Bilt, the Netherlands
[2]BeZero Carbon, London, United Kingdom
[3]Technical University Dresden, Germany





**Abstract**
Fires are a key component of the global carbon cycle and humans are changing their
characteristics. Fire emission monitoring is important to keep track of those changes and
TROPOMI satellite observations of tropospheric nitrogen dioxide, carbon monoxide and the
absorbing aerosol index can be used to quantify and verify the accuracy and precision of global
wildfire emission estimates on a daily basis. Here we use TROPOMI observations to evaluate
a new fire emission database based on Global Fire Atlas input for the Sense4Fire project (GFA-
S4F) and from the Copernicus Atmosphere Monitoring (CAMS) Global Fire Assimilation
System (GFAS) for a number of test regions worldwide representative of the most important
wildfire type environments. The main focus is on Amazon and Cerrado biomes (tropical rain
forests and deforestation) during August-September 2020, but analyses are also made for a
region in sub-Saharan Africa (savannah) as well as two regions in Siberia (steppe and boreal
forests/tundra). GFA-S4F and GFAS fire emissions are used as input for global atmospheric
composition model simulations based on IFS-COMPO, *i.e.* an extension of ECMWF's
Integrated Forecasting System (IFS) for simulating atmospheric composition. Comparing the
model output with the TROPOMI observations then provides an indirect check on the realism
of these emission estimates. Furthermore, for tropospheric nitrogen dioxide the IFS-COMPO
model simulations are also used to estimate the model sensitivity of tropospheric nitrogen
dioxide columns with respect to fire emission changes. This local relationship is used to
optimize the fire $NO_x$ emissions directly using the TROPOMI nitrogen dioxide observations.
The results reveal that for small fires emission nitrogen dioxide estimates are realistic on
average albeit with a large spread, *i.e.* for individual fires emissions can be significantly under
or overestimated regardless of emission database. However, for large fires nitrogen dioxide
emissions are systematically and largely overestimated in all four regions. The overestimation
can be an order of magnitude or even more. For area total nitrogen dioxide emissions this "large





fire bias" is of minor importance, *i.e.* total nitrogen dioxide emissions are dominated by small
fires. The GFA-S4F emission estimates were characterized by a larger positive bias for large
fire $NO_2$ emission cases compared to GFAS. The source of this bias is not well understood.
With optimized $NO_2$ emissions by direct adjustment of emission using TROPOMI nitrogen
dioxide observations the large positive bias can efficiently be resolved. Combined with an
update of soil $NO_x$ emissions – causing too low background $NO_x$ levels – a fairly good
agreement between IFS-COMPO and TROPOMI was reached.

Carbon monoxide was generally underestimated using GFAS emission (~50% on average

for the selected regions). Updating carbon monoxide emissions over the Amazon region by
incorporating more Sentinel satellite data (GFA-S4F) did reduce this fire CO bias significantly
(to ~25% on average).

Overall, the results show that TROPOMI data allows for systematically identifying

uncertainties and errors in satellite-data based fire emissions. The results also suggest that the
use of dynamic emission factors may further improve satellite based global emissions
inventories. In addition, the results also highlight that the use of TROPOMI data could be much
more detailed and refined towards assessing individual fires on a daily basis for better
understanding fire dynamics and to improve and diversify fire emission factors.



## 1. Introduction

Disturbance of vegetation by fire - anthropogenic or natural - is a major contributor to the amount of carbon (as carbon dioxide or methane) present in the atmosphere (Lasslop et al., 2019; Bowman et al., 2020; McLauchan et al., 2020). Vegetations fires are also important for the natural cycle of vegetation growth in many parts of the world and burning vegetation is a practice also used by humans in farming. The associated time scales can vary from several weeks to hundreds of years depending on vegetation type and speed of regrowth. Wildfire extremes and associated smoke can be disrupting to livelihoods as for example in Australia 2019 (Boer et al., 2020; Filkov et al., 2020) or the US West Coast 2020 (Higuera and Abatzoglou, 2021).

Satellite sensors can provide a number of key pieces of information to characterize vegetation fires (Chuvieco et al., 2019, 2020; Wooster et al., 2021). These include detection of thermal anomalies indicative of active fires, the energy released (fire radiative power or FRP), the loss of vegetation expressed as a change in surface reflectance indicative of burnt area and fire severity, or biomass loss, and observations of aerosols or atmospheric trace gases directly associated to, and traceable back to, fire events. Each individual dataset contains valuable fire information in itself but a greater understanding of the role of vegetation fires globally can be obtained by combining these datasets into one information system. Although several satellite-derived fire emission databases exist, there continues to be a need to develop additional validation methodologies and data products to advance our understanding of satellite-derived estimates of individual fire behavior (Andela et al., 2019, 2022; Andreae, 2019).

Earth observation can also help in constraining fire emission estimates. In particular carbon monoxide (CO) has been used for evaluation of fire emission estimates with various techniques, including formal emission inversions (Hooghiemstra et al., 2011; Yin et al., 2015),



mass budget analyses (Huijnen et al., 2016), Gaussian plume modelling (Adams et al., 2019)
as well as estimating fire $CO_2$ emissions using carbon monoxide as a proxy (Peiro et al., 2022).
Likewise aerosol and formaldehyde (HCHO) observations have been used as fire emission
proxies (Petrenko et al., 2012; Konovalov et al., 2014; Stavrakou et al., 2015; Bauwens et al.,
2016). However, in any of these methods the estimated carbon emissions are subject to
uncertainties in bottom-up emission estimates due to emission factors, the dynamics of the
emission process in the atmosphere and tracer lifetime. Limitations to data quality and the
spatio-temporal coverage of satellites further hamper in depth analysis of fire emissions to
larger regional to continental scales for many studies and trace gases (Alvarado et al., 2011;
Mebust et al., 2011, 2013, 2014; Young and Paton-Walsh, 2011; Castellanos et al., 2014;
Schreier et al., 2014; Tanimoto et al., 2015; Whitburn et al., 2015; Sitnov and Mokhov, 2017;
Lee et al., 2019; Adams et al., 2019; Lin et al., 2020).
The launch of the TROPOMI instrument on board of Polar orbiting Sentinel-5p satellite in
October 2017, with at that time unprecedented spatial resolution, data accuracy and precision,
has opened up a whole new range of possibilities for monitoring and studying fires. Several
research papers have been published in recent years exploring the use of TROPOMI CO and
$NO_2$ often in conjunction with FRP data from other satellites (Li et al., 2020; Griffin et al.,
2021; 2023; Jin et al., 2021; van der Velde et al., 2021; Stockwell et al., 2022; Wan et al., 2023;
see the Appendix for a brief summary of all these papers). These studies highlight the potential
of using TROPOMI data for assessing fire emissions. However, they also all note that their
studies are only first exploratory steps using TROPOMI and that more research is needed and
warranted while approaches could be expanded, extended and refined.
The ESA Sense4Fire project (S4F) explores the suite of the Sentinel satellite instruments
using a novel synergetic approach to derived global fire emissions based on the characterization



of individual fires and their behavior, eventually to better constrain total carbon emissions and
emission factors. Atmospheric chemical composition modelling is used as an interface between
Sentinel 2 and sentinel 3 based emissions *vs.* TROPOMI observations.
The objective of this study is to evaluate daily emission estimates of $NO_2$ and CO from the
Global Fire Atlas (GFA-S4F) and the Global Fire Assimilation System (GFAS) by using them
as input for atmospheric chemistry model simulations. The model results are compared with
TROPOMI observations of $NO_2$ and CO to assess the realism of these emission estimates. The
method described above is an indirect validation method in which the atmospheric composition
modelling results act as interface between the emission estimates and the TROPOMI data. We
therefore also apply an innovative approach for further updating and improving the emission
estimates that makes more direct use of TROPOMI observations.
**2 Data and methods**
**2.1 TROPOMI data**
The Sentinel-5 precursor satellite, launched on 13 Oct. 2017 in an ascending sun-
synchronous polar orbit, with an equator crossing at about 13:30 local time, carries the
TROPOspheric Monitoring Instrument (TROPOMI; Veefkind et al., 2012). Sentinel-5p is one
of the Sentinel satellites of the European Copernicus Program dedicated to monitoring
atmospheric composition. TROPOMI is a spectrometer that provides measurements in four
channels – ultraviolet (UV), visible (VIS), near infrared (NIR) and shortwave infrared (SWIR)
- of several atmospheric trace gases including $NO_2$ and CO and of cloud and aerosol
properties. The TROPOMI instrument is unique in several ways because it combines near-daily
global coverage with a wide spectral range, UV/VIS/NIR foot prints of $3.5 \times 5.5$ km$^2$ at nadir



$(3.5×7 \text{ km}^2$ before 6 August 2019), SWIR footprints of $5.5×7 \text{ km}^2$ at nadir $(7×7 \text{ km}^2$ before 6
August 2019) and a very large signal-to-noise ratio.

### 2.1.1 Tropospheric nitrogen dioxide ($NO_2$)

In this paper we use the TROPOMI $NO_2$ offline data from data processor version 2.3.1 and
algorithm version 1.5.0. The operational TROPOMI $NO_2$ product is described in van Geffen et
al. (2020). Detailed information can be found in the Product README File (PRF; Eskes and
Eichmann, 2021), the Product User Manual (PUM; Eskes et al., 2022) and the Algorithm
Theoretical Basis Document (ATBD; van Geffen et al., 2021).
Validation of TROPOMI tropospheric $NO_2$ columns for the biomass burning regions that
S4F focuses on is missing due lack of ground-based stations in those areas. The general
validation results for comparison with ground-based data (Verhoelst et al., 2021; Lambert et
al., 2023) indicate a negative bias for the tropospheric column data with a median value of 28%
with a range of 13% for rather clean locations to 40% over extremely polluted sites. The largest
differences occur during winter at higher latitudes (van Geffen et al., 2022). Note that these
biases fall (well) within the mission requirement of less than 50% bias. On the other hand,
given the lack of validation sites in the areas of interest of this paper – and in particular in the
tropical rain forest Amazon region and the south-of-the-equator African Savannah region it is
unclear how large TROPOMI tropospheric $NO_2$ columns biases are in those regions and
whether the validation would improve with the updated algorithm.

### 2.1.2 Carbon Monoxide (CO)

The TROPOMI CO total column retrieval algorithm derives data in the 2315–2338 nm
spectral range of the SWIR part of the solar spectrum and retrieves the CO values for clear-sky
conditions over land and low clouds over the ocean (Borsdorff et al., 2014; Landgraf et al.,



2016; Schneising et al., 2019). TROPOMI CO measurements are sensitive to the integrated
amount of CO along the light path, including the contribution of the planetary boundary layer,
making them particularly suitable for detecting surface sources of CO.

The operational TROPOMI CO retrieval deploys a profile scaling approach where a CO

reference profile is scaled to fit the TROPOMI reflectance measurements. For this, global,
monthly averaged vertical CO *a priori* profiles are used from the chemical transport model
TM5 (Krol et al., 2005). The forward calculation of the TROPOMI spectral measurements
account for light scattering by clouds and aerosols in the atmosphere and thus simultaneously
retrieves trace gas columns and effective parameters describing the cloud contamination of the
measurements (height scattering layer, scattering optical thickness) as demonstrated by Vidot
et al. (2012).

In this paper we use the TROPOMI CO offline data from data processor version 1.3.2 and

algorithm version 1.2.0. As recommended in the TROPOMI README file (Landgraf et al.,
2022a) and the product user manual (PUM; Apituley et al., 2022), we only use data with quality
assurance values (qa_values) larger than 0.5. More details about the algorithm can be found in
the Algorithm Theoretical Basis Document (ATBD; Landgraf et al., 2022b) that provides a
detailed reanalysis description of the implementation of the CO retrieval.

TROPOMI CO validation papers consistently report only small and mostly random biases

up to an order of magnitude smaller than the standard deviation of differences when compared
to ground-based observations, data and other satellite data. Correlations are generally high, and
biases are generally in the order of a few percent or less (Borsdorff et al., 2018, Martínez-
Alonso et al., 2020; Sha et al., 2021; Lambert et al., 2023). The differences fall well within the
TROPOMI mission requirements on accuracy (<15 %) and precision (<10 %) in CO total
columns. The data does suffer from striping issues and instrument effects in the area of the



South Atlantic Anomaly (SAA). For CO there is one validation site in the S4F Amazon area of
interest: Porto Velho. Validation results for Porto Velho reveal an excellent correlation (0.96)
and small bias (0.51%) in total CO columns of the offline data product (Lambert et al., 2023).
**2.1.3 Absorbing Aerosol Index (AAI)**
The AAI is a well-established satellite data product that has been produced for several
different satellite instruments spanning a period of more than 30 years. The AAI was first
calculated as a correction for the presence of aerosols in column ozone measurements made by
the TOMS instruments (Herman et al., 1997; Torres et al., 1998) because it was observed that
ozone values were too high in typical regions of aerosol emission and transport. The AAI is
based on spectral contrast in the ultraviolet spectral range for a given wavelength pair, where
the difference between the observed reflectance and the modeled clear-sky reflectance results
in a residual value. When this residual is positive, it indicates the presence of UV-absorbing
aerosols, like dust, smoke, or volcanic ash. Clouds yield near-zero residual values, and negative
residual values can be indicative of the presence of non-absorbing aerosols (*e.g.*, sulfate), as
shown by sensitivity studies of the AAI (de Graaf et al., 2005; Penning de Vries et al., 2009).
Unlike satellite-based aerosol optical thickness measurements, the AAI can also be calculated
in the presence of clouds so that daily global coverage is possible. This is ideal for tracking the
evolution of episodic aerosol plumes from dust outbreaks, volcanic eruptions, and biomass
burning. For this study, we use the TROPOMI AAI data for the wavelength pair 340–380 nm.
For more details about the TROPOMI AAI retrieval algorithm, see Stein-Zweers (2016). In
this paper we use the TROPOMI AAI offline data from data processor version 1.3.2 and
algorithm version 1.2.0.
**2.2 Methods**



We use fire emission data from two emissions inventories based on satellite data: the Global
Fire Atlas (GFA; Andela et al., 2017, 2019, 2022) emissions and the Global Fire Assimilation
System emissions (GFAS; Kaiser et al., 2012). We further use IFS-COMPO atmospheric
chemistry and transport model simulations (Huijnen et al., 2019; Williams et al., 2022),
TROPOMI data (Veefkind et al., 2012), and the innovative β-method for updating emissions
based on TROPOMI data (Lamsal et al., 2011; Castellanos et al., 2014). We also perform a
number of IFS-COMPO model experiments varying emissions or model processes/parameters
in order to better understand differences we find between IFS-COMPO and TROPOMI. The
particular experiments will be described in more detail later on (Table 2).
We perform an analysis in four study regions (see Appendix Fig. A1 and later Table 2) that
show a large variation of biomes and fire types. The main focus of this study is the
Amazon/Cerrado region; other regions are south-equatorial Africa savannahs, north Siberian
boreal forests and tundra, and central Siberian steppes. For the S4F project four 5°×5° areas
were selected to limit the high computation demand for calculating satellite data-based
emissions. However, given the IFS-COMPO resolution of 0.5°, a daily 5°×5° area would
frequently yield too little data for meaningful statistics. Hence why for this study we expanded
the coverage of the four regions (see later Table 2) to derive sufficient daily comparison data
of IFS-COMPO with TROPOMI data for meaningful statistics. Note that for the
Amazon/Cerrado we will refer to both the smaller and larger region, also to accommodate
future S4F research and publications.

### 215    2.2.1 Global Fire Atlas - based emissions

The Global Fire Atlas approach tracks individual fire events-based Moderate Resolution
Imaging Spectroradiometer (MODIS) burned area (Andela et al., 2019) or Visible Infrared
Imaging Radiometer Suite (VIIRS) active fire data (Andela et al., 2022). The VIIRS-based



method was developed to fill the need for a near–real-time approach for tracking contributions
from deforestation, forest, agricultural, and savanna fires to burned area and carbon emissions.
The approach was applied to the Amazon and Cerrado region, defined as the area 25°S-EQ,
85°W-30°W, for the years 2019 and 2020 although here we only will focus on emissions for
August and September 2020. Here we apply emissions factors derived from Andreae et al.
(2019) to translate carbon emissions to $NO_x$ and CO emissions for each fire type. The following
emissions factors (grams trace gas per kg matter burned) were used for grasslands and savanna
fires (69.2 and 2.49 g $kg^{-1}$), small clearing and agricultural fires (102 and 3.11 g $kg^{-1}$), forest
fires (98 and 1.94 g $kg^{-1}$), and deforestation fires (99 and 4.63 g $kg^{-1}$) for $NO_x$ (as NO) and CO,
respectively.
**2.2.2 GFAS fire emissions**
The Global Fire Assimilation System (GFAS; Kaiser et al., 2012) estimates dry matter
combustion rates by multiplying FRP and biome-specific emission factors. The global
distribution of FRP observations is obtained from the MODIS instruments on board the Terra
and Aqua satellites and are then assimilated into the GFAS system. The gaps in FRP
observations, which are mostly due to cloud cover and spurious FRP observations of volcanoes,
gas flares, and other industrial activity, are corrected or filtered in the GFAS system. Eight
biome-specific emission factors are used based on linear regressions between the GFAS FRP
and the dry matter combustion rate of Global Fire Emission Database (GFED) version 3.1 in
each biome (see later Table 2 and Fig. 3 in Kaiser et al. (2012)). The biomass burning emission
of a given species is then calculated by multiplying the dry matter combustion rate with an
emission factor of that species.
**2.2.3 IFS-COMPO**



As outlined in the introduction this study uses an atmospheric chemical composition model

(IFS-COMPO, previously known as "C-IFS" (Flemming et al., 2015)) as an interface between

Sentinel 2 and sentinel 3 based emissions and TROPOMI observations. IFS-COMPO is an

extended version of ECMWF's Integrated Forecasting System that was developed as part of

the global component of CAMS which includes modeling and assimilation of atmospheric

composition (aerosols, trace gases and greenhouse gases). Here we use a version of IFS-

COMPO which is set to simulate tropospheric chemistry and aerosol but excluding data

assimilation of atmospheric composition (see Appendix).

IFS-COMPO is run at a horizontal resolution of T511 (approximately 40 km grid cell), with

137 vertical levels and a time step of 900s. This default configuration of IFS-COMPO uses

CAMS-GLOB-ANT v5.3 anthropogenic emissions (Soulie et al., 2023), together with CAMS-

GLOB-BIO v3.1 biogenic emissions, and soil $NO_x$ emissions based on POET. The GFASv1.4

emissions, with updated emission factors for CO and NO, are applied globally. A series of

sensitivity experiments have been conducted, primarily testing the sensitivity in the fire input

emissions (Tables 1 and 2). To compare TROPOMI observations with IFS-COMPO output we

take into account all relevant aspects that are required when matching observation data to

model data including averaging kernels. Only TROPOMI observations with quality assurance

threshold above 0.75 are used, as recommended by the $NO_2$ product user manual. This concerns

observations with cloud radiance fraction of less than 0.5 and excludes retrievals with ground

pixels covered with snow/ice, as well as problematic retrievals.

The model fields are interpolated in time to match with local overpass time of TROPOMI

and the averaging kernel is applied to the model $NO_2$ profile. The collocated model-observation

pairs are gridded on a common 0.5°×0.5° output field (or different resolution, which is

configuration setting), and only written to output files if a threshold coverage of 50% of the



grid cell is reached. The averaging is done by an area-weighted approach, hence taking into
account the area of the TROPOMI-pixel that is within the model grid box (Douros et al., 2023).

Similar to the evaluation of IFS-COMPO NO$_2$, we use TROPOMI observations of CO total

columns to evaluate model CO columns, selecting observations with quality assurance
threshold above 0.5. The model total column fields are interpolated in time to match with local
overpass time of TROPOMI. The same grid for collocation is used as was adopted for the
evaluation against TROPOMI NO$_2$, again only grid cells with a threshold coverage of 50% are
used. Also the area averaging is the same as done for NO$_2$.

Because the TROPOMI CO column data is nearly uniformly sensitive to height (Borsdorff

et al., 2014) we will assume for comparison with IFS-COMPO that TROPOMI CO column
data represents a true vertical column so that no weighting or sensitivity correction on IFS-
COMPO CO data needs to be applied.
**2.2.4 β-method**

The basic approach followed in this paper is to use the IFS-COMPO model as

"intermediate" between the fire emission databases based on GFA or the GFAS emissions
database on the one end and the TROPOMI observations on the other end.

The IFS-COMPO model, however, also allows for applying a different approach to use

TROPOMI observations to modify and update model emissions by using the model simulations
to derive the local relationship between emissions and satellite measurements (Lamsal et al.,
2011; Castellanos et al., 2014). Although models like IFS-COMPO may simulate incorrect
trace gas amounts due to errors in fire emission estimates, they are capable of realistically
simulating changes in column amounts caused by changes in emissions. By performing a
baseline model simulation and a "perturbed emission" simulation, a local column-emission



sensitivity parameter β can be derived as function of space and time that connects changes in
column amounts ($\Delta TCNO_2$) to changes in emissions ($\Delta E$):
$$\frac{\Delta E}{E} = \beta\ \frac{\Delta TCNO_2}{TCNO_2}$$
Then, differences in measured and modelled columns can be converted into differences in
emissions relative to the baseline emissions by multiplication with the β parameter yielding an
updated TROPOMI-based emission estimate. Here, we use the β-method to assess to what
extent the prior fire NOx emission databases can be updated with this method to close the gap
between model simulations of fire $NO_x$ emission plumes and TROPOMI observations of $NO_2$.
For this we run sensitivity experiments where we scale down the prior fire emissions (either
GFAS or GFA-S4F) by 20%, and use the resulting change in tropospheric $NO_2$ columns to
compute local and daily varying β values.
The β value is determined by local atmospheric chemistry conditions and background $NO_x$
emissions in IFS-COMPO. A β value of one (1.0) indicates that the relative change in emissions
corresponds with a similar relative change in tropospheric $NO_2$ columns. β values < 1.0
indicating relatively low sensitivity of fire emission changes to column changes (β values > 1.0
indicative of the opposite). Very small β-values indicate limited sensitivity of emissions to
changes in column values, very large β-values indicate high sensitivity of emissions to small
changes in column values. Hence why β-values close to 1.0 are preferred and why we also limit
β-values to the 0.25-4.0 range.
To ensure that the β field only relate to fire emissions we additionally apply a filtering
procedure to exclude instantaneous values of β when prior emissions are smaller than 0.1 mg
$m^{-2}$ $d^{-1}$ and model $NO_2$ columns are smaller than $2 \times 10^{15}$ molecules $cm^{-2}$, and additionally take





the local median value of β computed based a two-month time series (August-September 2020). Over the Amazon, more than 97% of the median β values fall within the 0.5-2.0 range with 60% within the 1.0-1.5 range (see Appendix Fig. A2). Note that on average β-values get closer to 1.0 for larger tropospheric $NO_2$ column values indicating that the larger emissions, the more linear and straightforward the relation between emissions and $NO_2$ column values.

Applying the β-method comes with a number of caveats and limitations. It does require prior emissions to be present for a given grid location in the IFS-COMPO model if the emissions are to be updated. This is a different approach from for example emission inversion methods that do not require any *a priori* information (Mijling et al., 2013; Ding et al., 2017). Also, given the differences in spatial resolution between IFS-COMPO, TROPOMI data and the emission databases need to be kept in mind. The β-method only allows to translate TROPOMI column enhancements into emission optimization within the (coarser) model grid resolution, which is valid for trace gases with sufficiently short lifetime such as $NO_2$. Furthermore, the β-method assumes that column amounts and emissions vary linearly which may not always be the case. Hence why β values close to a value of 1.0 are preferred and large changes in emissions far beyond the 20% model emission perturbation should be carefully considered. In principle non-linear relationships between column amounts and emissions could be overcome by applying the method iteratively albeit at the cost of requiring more model simulations and thus time. Nevertheless, once the model simulations have been performed the β-method provides a straightforward method to use TROPOMI data for a rapid first order update of *prior* emissions.

**3. Results**

**3.1 Amazon**



Fig. 1 show a SUOMI-NPP VIIRS image for 11 September 2020 over the selected
Sense4Fire Amazon region as well as the larger Amazon region. They reveal a pattern typical
for this Amazon region during this time of the year. There are widespread fires and smoke
plumes visible over regions where deforestation is taking place. There is some shallow
convection present, but weather conditions are mainly dry. Smoke from the fires covers a large
region in the Amazon (Fig. 1, lower panel), resulting in accumulation of pollution with the
Andes mountains to the west acting as a barrier for transport of pollution out of the region.



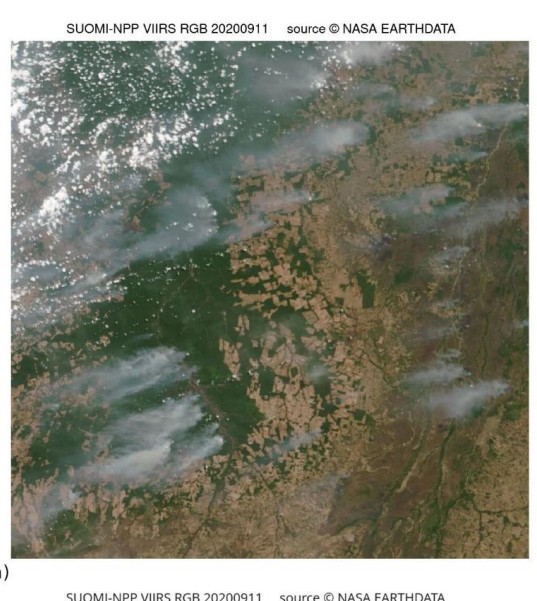

(a)

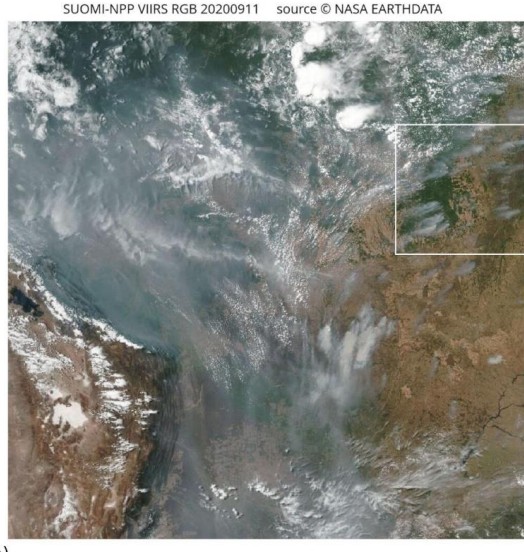

(b)

**Figure 1.** (a) SUOMI-NPP VIIRS RGB image on 11 September 2020 over the Amazon region

between 50°W-55°W, 9°S-14°S. Image obtained from NASA WorldView based on the python

script by Brian Blaylock (Univ. Utah, 2015); (b) as Fig. 1a but for 50°W-70°W, 5°S-25°S. The

area of Fig. 1a is denoted by the white square.



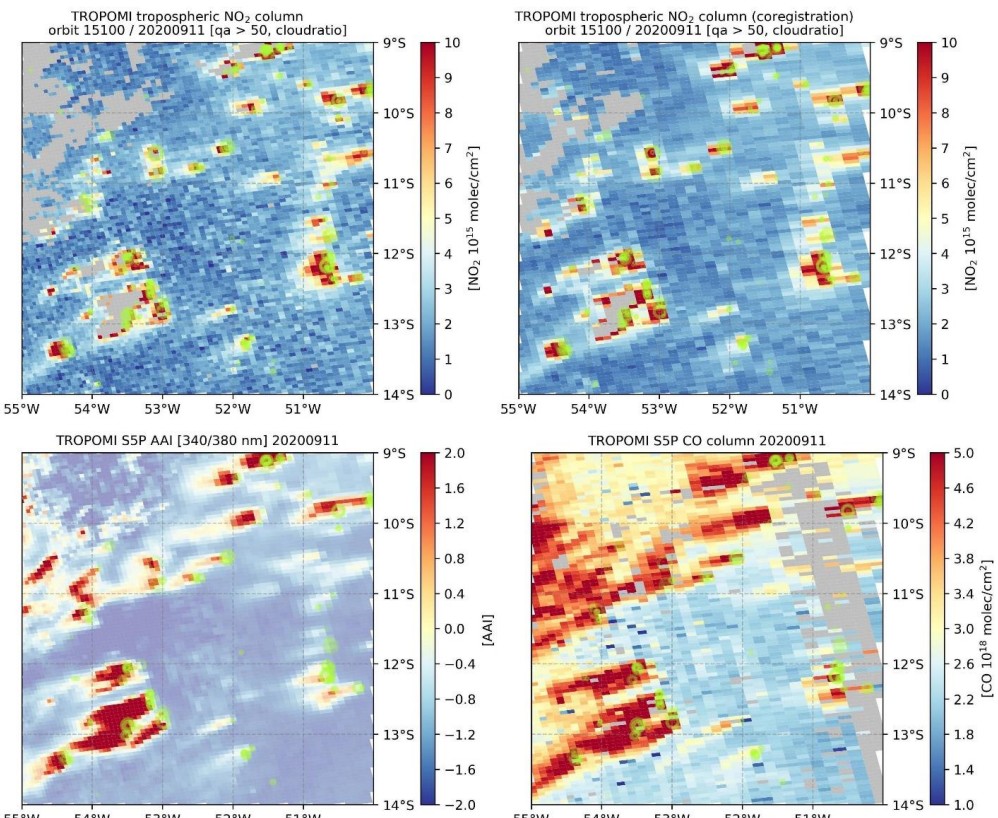

**Figure 2.** TROPOMI measurements of tropospheric NO₂ column (original resolution a regridded to the CO grid), the CO total column and the AAI on 11 September 2020 for the region shown in Fig. 1 (upper panel). The open green circles depict coincident NPP-VIIRS FRP measurements with the radius of the circles representing the magnitude of the FRP (arbitrary unit). Only measurements with TROPOMI NO₂ quality flag values > 0.5 are shown. The regridding of tropospheric NO₂ column was done using a python based coregistration algorithm (M. Sneep, KNMI, personal communication, 2023; available on request). Pixels in which the cloud pressure was within 4% of the surface pressure were also included to in particular allow for pixels with enhanced NO₂ over low altitude smoke, following van der A et al. (2020).





Fig. 2 shows TROPOMI observations over the same region as Fig. 1 (upper panel). Many
fire emission plumes can be discerned in tropospheric $NO_2$, CO and the AAI data. On close
inspection the plumes generally emanate from where SUOMI-NPP FRP indicates fire events
(wind direction was east-north-east). Close to fires tropospheric $NO_2$ is enhanced which rapidly
drop to background column values typically within 5-10 TROPOMI pixels, approximately 25-
50 km distance and reflecting the relative short lifetime of tropospheric $NO_2$ of a few hours at
maximum in this moist and sunlit region. For CO and the AAI the plumes extend much further
reflecting the much longer lifetime of both parameters relative to tropospheric $NO_2$. On the
timescales of plume advection (hours to a day) CO and AAI act as passive tracers with plume
variations dominated by turbulence and dispersion. For tropospheric $NO_2$ photochemical
equilibrium and chemical loss also plays a role. The tropospheric $NO_2$ data also reveal that for
large AAI values and thus optically thick smoke no accurate tropospheric $NO_2$ column values
(low quality flag value) could be retrieved even though total $NO_2$ data do show enhanced total
$NO_2$ over the smoke (not shown). Thick smoke is considered a cloud in the tropospheric $NO_2$
retrieval algorithm, hence the low data quality flag value.
Fig. 3 show the 2-D probability distributions of daily TROPOMI $NO_2$, CO, and AAI for
the large Amazon region of Fig. 1 for the entire month of September. As expected based on
Fig. 2, CO and AAI correlate well whereas tropospheric $NO_2$ hardly correlates with either.



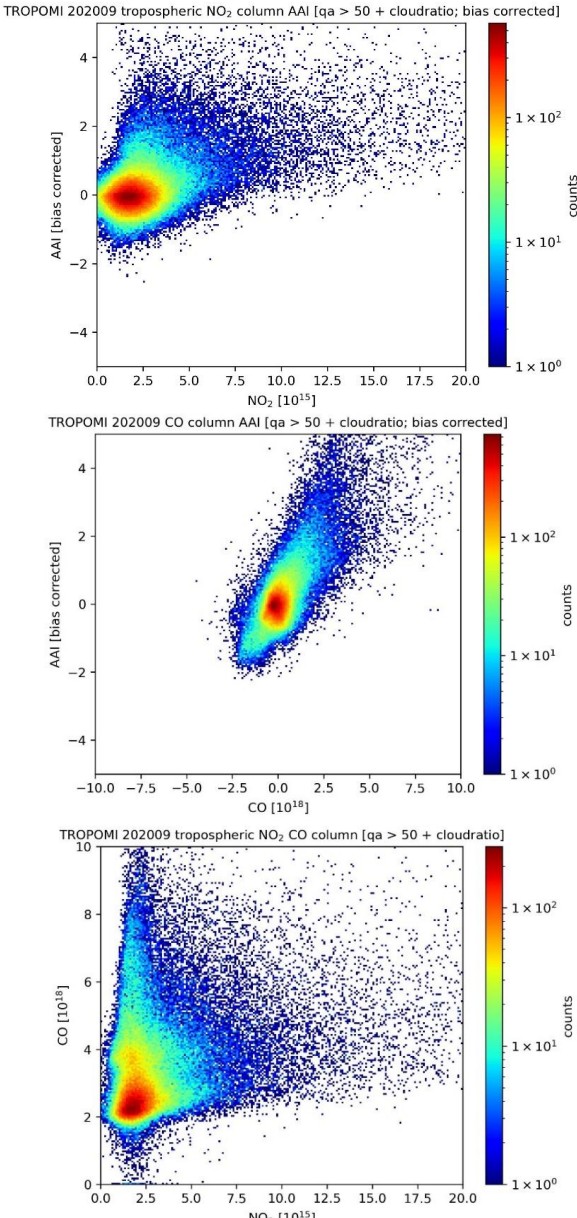

Comparing these distributions for different months for the same region in 2019 and 2020 shows

that August and September are the dominant Amazon fire months (see Appendix Fig. A4).

**Figure 3.** 2D histogram of daily September 2020 TROPOMI data for AAI and CO (upper

panel), CO and NO$_2$ (middle panel) and AAI and NO$_2$ (lower panel) for the same region as



shown in Fig. 2. For the upper panel the AAI and CO data are biases corrected on a daily basis,
*i.e.* each day the median value of the daily probability distribution is subtracted. See Appendix
Fig. A3 for the same figure without the bias correction. $NO_2$ data is not bias corrected (middle
and lower panel) and CO data for the lower panel is also not corrected, see Appendix Fig. A3
for the same figure with the CO bias correction as applied for the upper panel of Fig. 3 here.
AAI data is unitless, CO data is in $10^{18}$ molecules $cm^{-2}$, $NO_2$ data is in $10^{15}$ molecules $cm^{-2}$



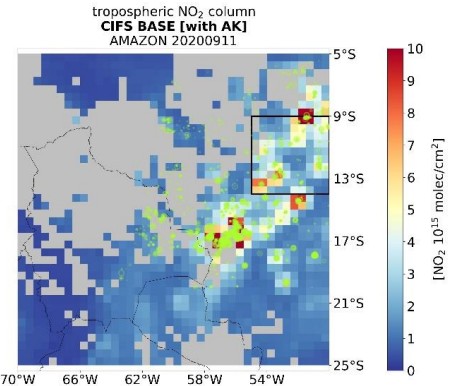

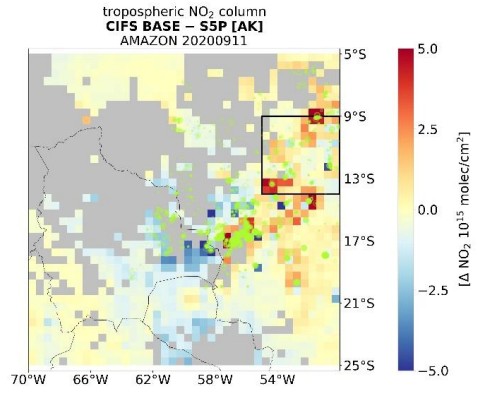

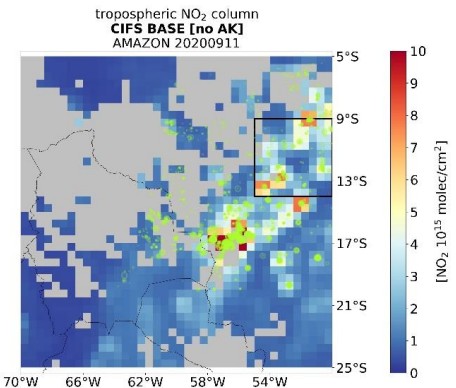

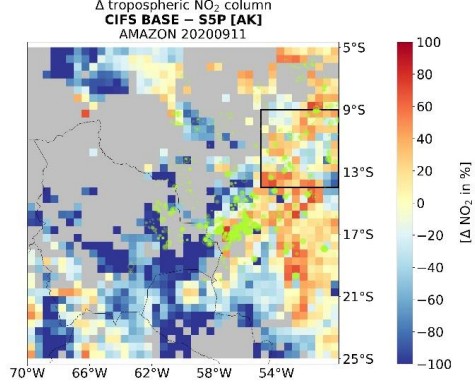

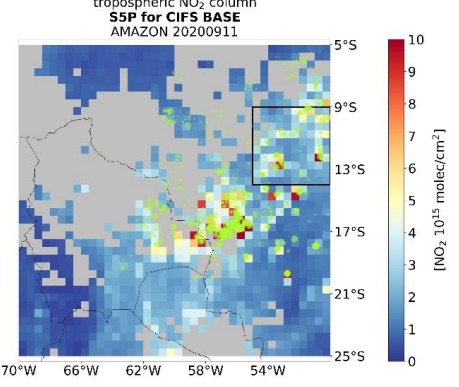

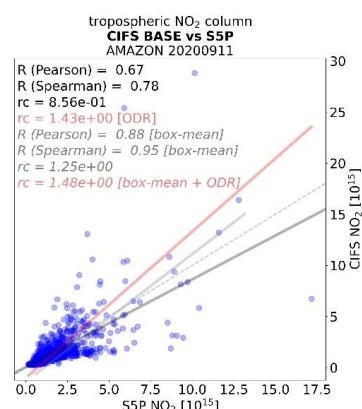




**Figure 4.** Tropospheric $NO_2$ columns for the larger Amazon region (Fig. 1, lower panel) for the IFS-COMPO simulation using GFAS emissions and applying the corresponding TROPOMI $NO_2$ averaging kernel (upper left) and without applying the averaging kernel (middle left) with the corresponding IFS-COMPO grid averaged tropospheric $NO_2$ observations (lower left). Differences between IFS-COMPO and TROPOMI in the upper right panel (absolute) and middle right panel (relative) and corresponding scatter plot and associated statistics (lower right panel). The small region from Fig. 1 are indicated by the black box. SUOMI-NPP VIIRS FRP are in the bright green circles as in Fig. 2. The statistics in the lower right plot display the correlation coefficients for all data points (Pearson's and Spearman's); the corresponding ordinary linear regression and orthogonal distance regression (ODR; in red); and the same statistics but for observations averaged in twenty equally distributed TROPOMI data intervals ("box mean"; in grey and in italics).

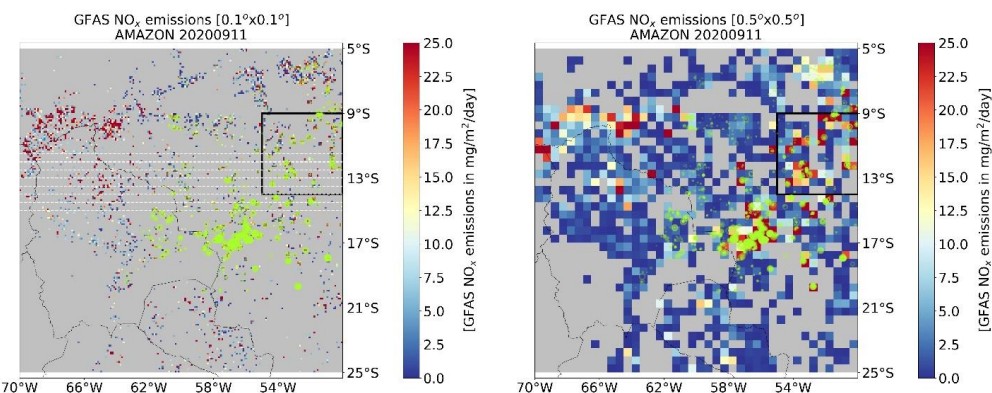

**Figure 5.** GFAS emissions at 0.1°×0.1° (lower plot) and summed at 0.5°×0.5° right plot for the large Amazon region as shown in Fig. 1 (lower panel).

Fig. 4 shows a comparison of the IFS-COMPO model simulation configured with its default settings as also operated in CAMS (GFAS emissions displayed in Fig. 5) and TROPOMI observed tropospheric $NO_2$ columns for the same day and large Amazon region as in Fig. 1.





There is a reasonable correlation between observed and modeled tropospheric $NO_2$ columns
(0.67 and 0.78 for respectively $R^{PEARSON}$ and $R^{SPEARMAN}$) but the associated orthogonal linear
regression coefficient (RC) is significantly larger than one (1.43). Averaging tropospheric $NO_2$
column data to account for the spread in the distribution of data points improves the correlations
(0.88/0.95) but the large RC remains (1.8). For this day there is a cluster of fires and plumes
south and southwest (14°S-18°S, 60°W-50°W) of the smaller Amazon region of Fig. 1 where
all IFS-COMPO values overestimate tropospheric $NO_2$. On the other end, outside of major fire
areas IFS-COMPO tends to underestimate observed tropospheric $NO_2$, possibly linked to soil
$NO_x$ emissions, which will be discussed later. The difference plots show that locally differences
between model simulations and observations can easily be 100% or more.
To explore the presence of systematic biases all collocated daily data for August and
September 2020 for the larger Amazon region of Fig. 1 were combined into 2D histograms
shown in Fig. 6. The statistics reveal a fair correlation of 0.47 and 0.72 with a relatively small
uncertainty range and a RC of almost 0.80. Averaging data similar as done in Fig. 4 improves
the comparison with correlations of 0.88 and 0.97 and a larger orthogonal linear RC of 0.90.
More or less similar numbers are found for the smaller Amazon region. Visual inspection of
Fig. 6, however, reveals that there is a significant model bias for large tropospheric $NO_2$
columns, *i.e.* IFS-COMPO overestimates tropospheric $NO_2$ columns and differences can be
multiple factors up to an order of magnitude or more. The opposite, IFS-COMPO more than
an order of magnitude smaller than TROPOMI hardly occurs (see Appendix Table A1).
Strongly enhanced IFS-COMPO tropospheric $NO_2$ column values in this region are
predominantly associated with fire emissions rather than emissions from other sources. Hence,
the IFS-COMPO "large tropospheric $NO_2$ column" bias is thus associated with larger fire
emissions.



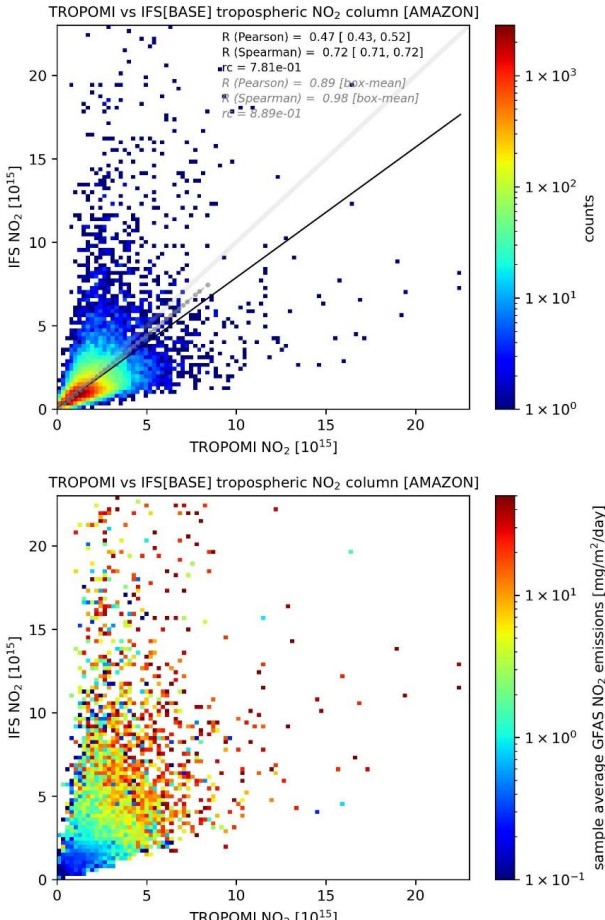


**Figure 6.** 2D histogram of TROPOMI observed and IFS-COMPO simulated tropospheric $NO_2$

columns for daily observations throughout August and September 2020 for the larger amazon

region (Fig. 1). The black line indicates the regression coefficient for all data, the grey line

("box-mean") when IFS-COMPO data are averaged within twenty TROPOMI bins (only with

more than ten data points in a particular TROPOMI bin). The lower panel displays the same

distribution as in the upper panel but color coded according to the average GFAS $NO_2$ emission.

Note that distributions between both differ slightly as occasionally for a TROPOMI - IFS-

COMPO comparison the corresponding GFAS emissions are zero (used in the lower plot).



To further explore this IFS-COMPO "large tropospheric $NO_2$ column" bias a number of
IFS-COMPO model experiments were performed (see Tables 1 and 2 and Appendix Table A1).
Replacing the GFAS fire $NO_x$ emissions in IFS-COMPO with GFA $NO_x$ emissions
(experiments GFA, GFA.IFSCYCLE, GFA.SOIL; Appendix Fig. A5) resulted in larger
observed differences. Although correlations remained similar, lower regression coefficients
indicate larger differences between observations and model outcomes.
Another test was to put a hard cap on $NO_x$ emissions (see Appendix Fig. A6). For
BASE.CAP0.1 the $NO_x$ emission cap was set at $1 \cdot 10^{-10}$ kg m$^{-2}$ s$^{-1}$ (~0.1 mg m$^{-2}$ d$^{-1}$), for
BASE.CAP0.3 this was $3 \cdot 10^{-10}$ kg m$^{-2}$ s$^{-1}$ (~0.3 mg m$^{-2}$ d$^{-1}$; both with GFAS emissions) while
for GFA.CAP0.3 the same $3 \cdot 10^{-10}$ kg m$^{-2}$ s$^{-1}$ emission cap was used but for the GFA-S4F
emissions. Although capping the emissions obviously reduces the presence of the "large
tropospheric $NO_2$ column" bias it did not significantly improve the correlation and even
worsened the regression coefficients.
An alternative approach was to directly use TROPOMI $NO_2$ column data to constrain
emissions (BASE.BETA, GFA.BETA; Appendix Figs. A6 and A7) using the β-method that
was introduced in section 2.5. This indeed improves the regression results and especially for
the GFA-S4F emissions the "large tropospheric $NO_2$ column" bias is strongly reduced (number
of strongly deviating IFS-COMPO pixels reduced by an order of magnitude) with orthogonal
linear regression coefficients much more in line with the baseline results for GFAS (BASE):
for instance, the spearman correlation coefficient is 0.75 for both experiments with the β-
method applied, where it was 0.60 for the reference GFAS experiment (Table 1). The fact that
overall very similar results are achieved with the β-method, independent of the prior emissions,
gives confidence in its procedure which to first order is independent of the prior emissions.
Nevertheless, these experiments fail in improving the RC (RC = 0.729, 0.739 for experiments



461 with β-method, and RC = 0.764 for the BASE experiment), pointing at a common negative

462 model bias largely independent of fire emissions.

463 These results indicate that a large portion of the remaining negative model bias was not

464 affected and optimized during the fire $NO_x$ optimization procedure, and therefore is likely

465 attributed to other emissions than fires. In particular the soil $NO_x$ emissions in the default

466 configuration of IFS-COMPO CY48R1 are identified to be comparatively low. For that reason

467 a set of IFS-COMPO simulations was performed with updated soil $NO_x$ emissions, both

468 without, and with optimized fire $NO_x$ emissions (IFS-COMPO experiments GFA.SOIL and

469 GFA.BETA.SOIL see Table 1, Appendix Figs. A5 and A7). Solely updating the soil $NO_x$

470 emissions had a limited effect on the statistics (RC = 2.15 in GFA.SOIL *vs.* 2.11 in

471 GFA.IFSCYCLE), indicating that the biases due to fire $NO_x$ emissions in GFA-S4F are

472 dominating. But combined with β-optimization the statistics improved and especially the

473 orthogonal linear regression coefficients approached the value of one (RC = 0.953 in

474 GFA.BETA.SOIL, vs 0.939 in GFA.BETA).





**Figure 7.** Panels (A, B) as Fig. 6 but for the Africa region (10°E-30°E and 5°S-25°S). IFS-COMPO simulations with GFAS emissions and updated soil $NO_x$ emissions; panels (C, D) as panels (A, B) but with β-optimized GFAS emissions; panels (E, F) as panels (A, B) but for the Siberia tundra region (125°E-145°E, 55°N-75°N); panels (G, H) as panels (A, B) but with β-optimized GFAS emissions for the Siberia tundra region (125°E-145°E, 55°N-75°N).

**3.2 Other regions: sub-equatorial African savannahs, Siberian steppes and tundra**

The quality of GFAS fire $NO_x$ emissions, and the optimization based on the β-method was further explored for a selection of other regions (a sub-equatorial region in Africa, a Siberia tundra region and a Siberia steppe region, see Appendix Fig. A2). This choice was motivated by the very different vegetation types, soils characteristics and weather and climatological conditions of each region: sub-equatorial Africa fires are dominated by savannahs and arid shrublands; the Siberia tundra fires are dominated by wet evergreen forest and tundra vegetation; the Siberian steppe fires are dominated by vast grasslands. They therefore provide clues as to whether the agreement and discrepancies found for the Amazon/Cerrado region generally hold or are just a regional phenomenon. For these other regions we solely rely on the IFS-COMPO simulations with GFAS emissions and updated soil $NO_x$ emissions while comparing results with and without β-optimization. Table 2 summarized the results for these three regions.

For sub-equatorial Africa (Fig. 7 panels A – D), observed and modeled tropospheric $NO_2$ columns have a similar dynamical range, a similar spread and variability, and a similar dependence of larger tropospheric $NO_2$ columns over regions with larger emissions. However, the IFS-COMPO simulations significantly and systematically underestimate tropospheric $NO_2$ columns





| IFS run ID | region | IFS setup | emissions | all data R$^{PEARSON}$ [CI] / R$^{SPEARMAN}$ [CI] | all data RC | averaged data R$^{PEARSON}$ / R$^{SPEARMAN}$ | averaged data RC |
|---|---|---|---|---|---|---|---|
| **Nitrogen Dioxide (NO$_2$)** | | | | | | | |
| b2em GFA.SOIL | Africa | updated soil NO$_x$ | GFA-S4F | 0.62 [0.60, 0.64] / 0.77 [0.77, 0.77] | 0.376 | 0.93 / 0.96 | 0.270 |
| b2ew GFA.BETA.SOIL | Africa | updated soil NO$_x$ β-optimized emissions | GFA-S4F | 0.64 [0.62, 0.65] / 0.78 [0.78, 0.78] | 0.733 | 0.93 / 0.97 | 0.882 |
| b2em GFA.SOIL | Siberia-tundra | updated soil NO$_x$ | GFA-S4F | 0.40 [0.39, 0.41] / 0.46 [9,45, 0.47] | 0.385 | 0.84 / 0.79 | 0.297 |
| b2ew GFA.BETA.SOIL | Siberia-tundra | updated soil NO$_x$ β-optimized emissions | GFA-S4F | 0.43 [0.42, 0.44] / 0.46 [0.45, 0.47] | 0.365 | 0.85 / 0.79 | 0.271 |
| b2em GFA.SOIL | Siberia-steppe | updated soil NO$_x$ | GFA-S4F | 0.62 [0.61, 0.63] / 0.65 [0.64, 0.65] | 0.493 | 0.96 / 0.99 | 0.556 |
| b2ew GFA.BETA.SOIL | Siberia-steppe | updated soil NO$_x$ β-optimized emissions | GFA-S4F | 0.62 [0.61, 0.63] / 0.65 [0.64, 0.65] | 0.494 | 0.97 / 0.99 | 0.555 |
| **Carbon Monoxide (CO)** | | | | | | | |
| b2bd BASE | Amazon | | GFASv1.4 | 0.88 [0.88, 0.88] / 0.94 [0.94, 0.94] | 0.571 | 0.89 / 0.86 | 0.351 |
| b2bj GFA | Amazon | | GFA-S4F | 0.87 [0.87, 0.88] / 0.93 [0.93, 0.94] | 0.792 | 0.92 / 0.99 | 1.020 |
| b2bd BASE | Africa | | GFASv1.4 | 0.64 [0.61, 0.66] / 0.75 [0.74, 0.76] | 0.292 | 0.98 / 0.99 | 0.300 |
| b2bd BASE | Siberia-tundra | | GFASv1.4 | 0.74 [0.73, 0.74] / 0.83 [0.83, 0.83] | 0.694 | 0.93 / 0.97 | 0.428 |
| b2bd BASE | Siberia-steppe | | GFASv1.4 | 0.78 [0.78, 0.79] / 0.75 [0.75, 0.76] | 0.358 | 0.97 / 1.00 | 0.361 |






**Table 1.** Overview of statistics of the comparison of IFS-COMPO simulated and TROPOMI
observed tropospheric $NO_2$ column over the larger Amazon region (like Figs. 6-7-8). All
simulations used the CBM5 atmospheric chemistry scheme. Several simulations also make use
of a subgrid-scale emission plume chemistry-dispersion parameterization scheme (SGS) that
accounts for the fact that most emission plumes are significantly smaller than the CAMS/CIFS
grid size, and that plume chemistry thus is a subgrid scale process. Sensitivity tests revealed
that this subgrid scale parameterization had only minor impacts on simulated $NO_2$ and CO.
Box-mean data refers to the statistics of the average values. The four-character IDs refers to
the ECMWF supercomputer simulations and are included here for traceability purposes.





| IFS run ID | IFS setup | emissions | IFS version | $R^{PEARSON}$ [CI] / $R^{SPEARMAN}$ [CI] | RC | $R^{PEARSON}$ / $R^{SPEARMAN}$ | RC |
|---|---|---|---|---|---|---|---|
| b2bd BASE | "large Amazon region" | GFASv1.4 | CY47R3.1 | 0.47 [0.44, 0.52] / 0.72 [0.72, 0.73] | 0.798 | 0.88 / 0.97 | 0.900 |
| b2bd BASE | "small Amazon region" | GFASv1.4 | CY47R3.1 | 0.57 [0.53, 0.62] / 0.60 [0.58, 0.62] | 0.764 | 0.94 / 0.94 | 0.820 |
| b2by BASE.CAP0.1 | $NO_x$ emissions capped $1 \cdot 10^{-10}$ kg m$^{-2}$ s$^{-1}$ | GFASv1.4 | CY47R3.1 | 0.64 [0.63, 0.65] / 0.70 [0.69, 0.70] | 0.304 | 0.93 / 0.94 | 0.227 |
| b2c4 BASE.CAP0.3 | $NO_x$ emissions capped $3 \cdot 10^{-10}$ kg m$^{-2}$ s$^{-1}$ | GFASv1.4 | CY47R3.1 | 0.63 [0.62, 0.64] / 0.72 [0.71, 0.72] | 0.473 | 0.96 / 0.96 | 0.421 |
| b2bj GFA | | GFA-S4F | CY47R3.1 | 0.36 [0.34, 0.40] / 0.73 [0.72, 0.73] | 2.230 | 0.86 / 0.96 | 3.340 |
| b2c6 GFA.CAP0.3 | $NO_x$ emissions capped $3 \cdot 10^{-10}$ kg m$^{-2}$ s$^{-1}$ | GFA-S4F | CY47R3.1 | 0.59 [0.57, 0.61] / 0.74 [0.73, 0.74] | 0.623 | 0.94 / 0.94 | 0.577 |
| b2d3 GFA.IFSCYCLE | | GFA-S4F | CY48R1.0 SGS | 0.35 [0.32, 0.39] / 0.73 [0.73, 0.74] | 2.110 | 0.85 / 0.99 | 3.210 |
| b2dl BASE.BETA | β-optimized emissions | GFASv1.4 | CY48R1.0 SGS | 0.50 [0.44, 0.61] / 0.75 [0.74, 0.75] | 0.729 | 0.86 / 0.98 | 0.868 |
| b2dz GFA.BETA | β-optimized emissions | GFA-S4F | CY48R1.0 SGS | 0.41 [0.36, 0.49] / 0.75 [0.75, 0.76] | 0.739 | 0.97 / 0.98 | 0.752 |
| b2em GFA.SOIL | updated soil $NO_x$ | GFA-S4F | CY48R1.0 SGS | 0.36 [0.33, 0.40] / 0.75 [0.74, 0.75] | 2.150 | 0.86 / 0.99 | 3.230 |
| b2ew GFA.BETA.SOIL | updated soil $NO_x$ β-optimized emissions | GFA-S4F | CY48R1.0 SGS | 0.36 [0.31, 0.43] / 0.78 [0.77, 0.78] | 0.953 | 0.95 / 0.98 | 1.04 |






**Table 2.** As Table 1 but for baseline simulations and other regions for both tropospheric $NO_2$
and CO. All simulations used the CBM5 atmospheric chemistry scheme and the subgrid-scale
emission plume chemistry-dispersion parameterization (SGS, see caption Table 1).

* Amazon "large"    = 70°W - 50°W      25°S-5°S

* Amazon "small"    = 55°W - 50°W      14°S-9°S

* Africa             = 10°E -  30°E       25°S-5°S

* Siberia-tundra     = 125°E - 145°E    55°N-75°N

* Siberia-steppe     = 40°E -  60°E      40°N-60°N


which is opposite from the Amazon/Cerrado region. Updating GFAS emissions using the β-
optimization significantly improves the comparison, in particular the regression coefficient.
For Siberia the conditions are very different from those in the Amazon and Africa. First of
all, the dynamical range of tropospheric $NO_2$ columns is much smaller (compare Figs. 6 and
Fig. 7 panels E – H and Fig. 8 panels A - D) and there are fewer fires as evidenced by a limited
number of points outside of the main probability distribution. Especially for the Siberia Steppe
region fire emissions are very small. Although both Siberia regions show a tropospheric $NO_2$
column bias not dissimilar from those in Africa, applying the β-optimization does not result in
a large improvement unlike for the Amazon and Africa regions. Given that there are fewer fires
in Siberia in the particular period studied here, this may not be that surprising as there are not
many fire-affected regions and thus tropospheric $NO_2$ columns to β-optimize. Note that the
tropospheric $NO_2$ columns for Siberia (especially tundra) and fire $NO_2$ emissions are much
smaller than those for the Amazon/Cerrado and Africa. Which is unlike CO for which column
values and emissions are comparable (see next section, Table 2 and Appendix Fig. A8). This
reflects differences in fire characteristics: boreal vegetation is wetter and burning will be more



incomplete (more CO and smoke) and at much lower temperatures (less $NO_2$) (Andreae, 2019;
van Wees et al., 2022).

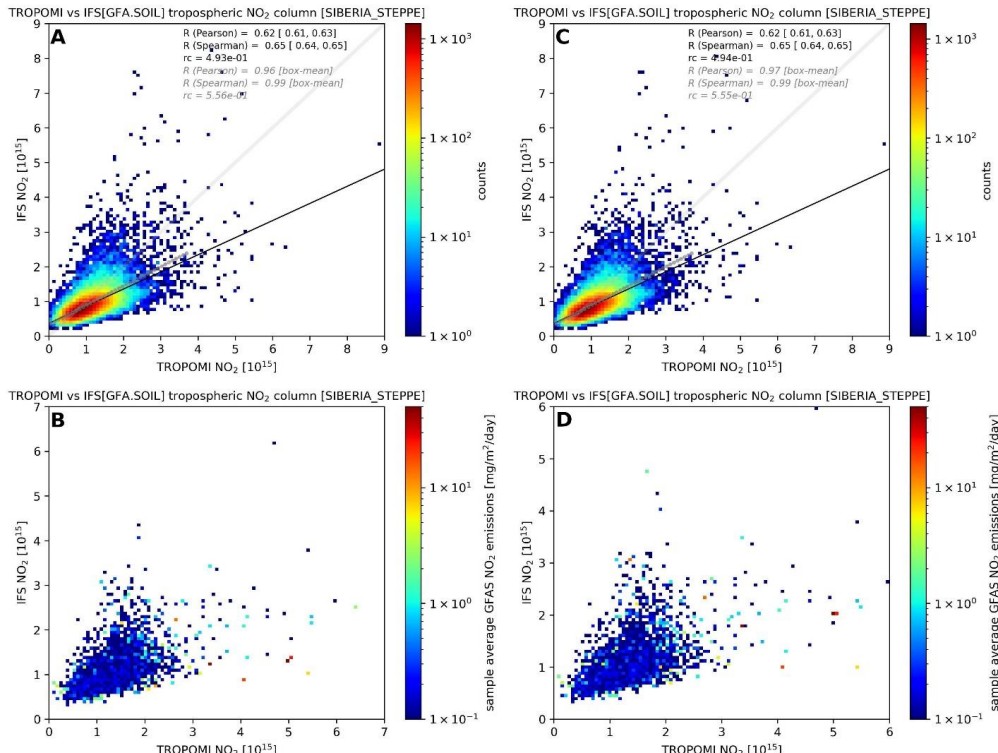


**Figure 8.** Panels (A, B) as Fig. 6 but for the S4F Siberia steppe region (40°E-60°E, 40°N-

60°N); panels (C, D) as panels (A, B) with β-optimized GFAS emissions but for the S4F

Siberia steppe region (40°E-60°E, 40°N-60°N)

**3.3 Carbon Monoxide**

Next we present the comparison of the IFS-COMPO simulations of CO – see further Table

2. In this case the number of IFS-COMPO tests was limited to comparing IFS-COMPO either
with GFAS or GFA CO emissions. Overall, for all regions there is a good correlation between
modeled and observed total CO columns, better than for $NO_2$. This likely reflects the simpler
chemistry and longer lifetime of CO, causing CO to vary on larger spatial scales that are easier





for IFS-COMPO to capture. For all regions IFS-COMPO nevertheless consistently and
significantly underestimates CO columns by 20-70%. However, and importantly, the
regression coefficient when using GFA-S4F emissions over the Amazon region significantly
improves the comparison to the level that the observations and model results compare very
well. A small bias may remain but given that the GFA-S4F CO emission data was only
available over the Amazon region and for example not the entire South American continent
and combined with the long CO lifetime the remaining small bias may simply be the results of
lack of updating emissions outside of the S4F Amazon region (missing advection of additional
CO outside of the area of interest). This suggests that despite the larger bias in terms of $NO_2$,
the prior emission estimates of dry-matter burned in GFA-S4F over the amazon are likely better
than GFAS, and the discrepancy with respect to TROPOMI $NO_2$ points rather at uncertainties
in the $NO_x$ emission factor.
**3.4 Time series**
Finally, a key question regarding the fire $NO_2$ emissions results discussed here is how much
in particular the "large tropospheric $NO_2$ column" bias really matters. To answer that question,
Fig. 9 shows the daily total $NO_2$ emissions for the Amazon region for four different emission
databases: GFAS, GFA-S4F and the β-optimized emissions for both.
The comparison first reveals that the temporal variability in $NO_2$ emissions for the Amazon
in GFAS and GFA-S4F are very comparable. There are some differences, but overall temporal
variability in emissions as well as the amplitude of emissions are similar. The second notable
result is that the β-optimization has a significant impact on in particular the GFA-S4F $NO_2$
emissions, and provides results that are very similar compared to the other estimates (GFAS
and GFAS β-optimized) in terms of temporal variability, while the area-and time-averaged
emission totals, quantified in terms of daily mean emissions, are overall reduced. It was shown



previously that the GFA-S4F $NO_2$ emissions significantly worsened "large tropospheric $NO_2$
column" bias. The β-optimization nevertheless can largely correct for this bias. This is a
valuable result as it allows - at least to first order – to independently provide an estimate of the
fire $NO_x$ emissions based on TROPOMI observations for evaluation and verification of bottom-
up emission databases. The impact of the "large tropospheric $NO_2$ column" bias on emission
totals nevertheless is rather small. Total $NO_x$ emissions differ on average by 10% or less. That
indicates that not only a small subset of larger fires which appear over-estimated in GFA-S4F
is important to match the emission totals, but also a larger, dominating subset of smaller fires,
with low $NO_x$ emissions, which may be under-estimated in GFA-S4F.

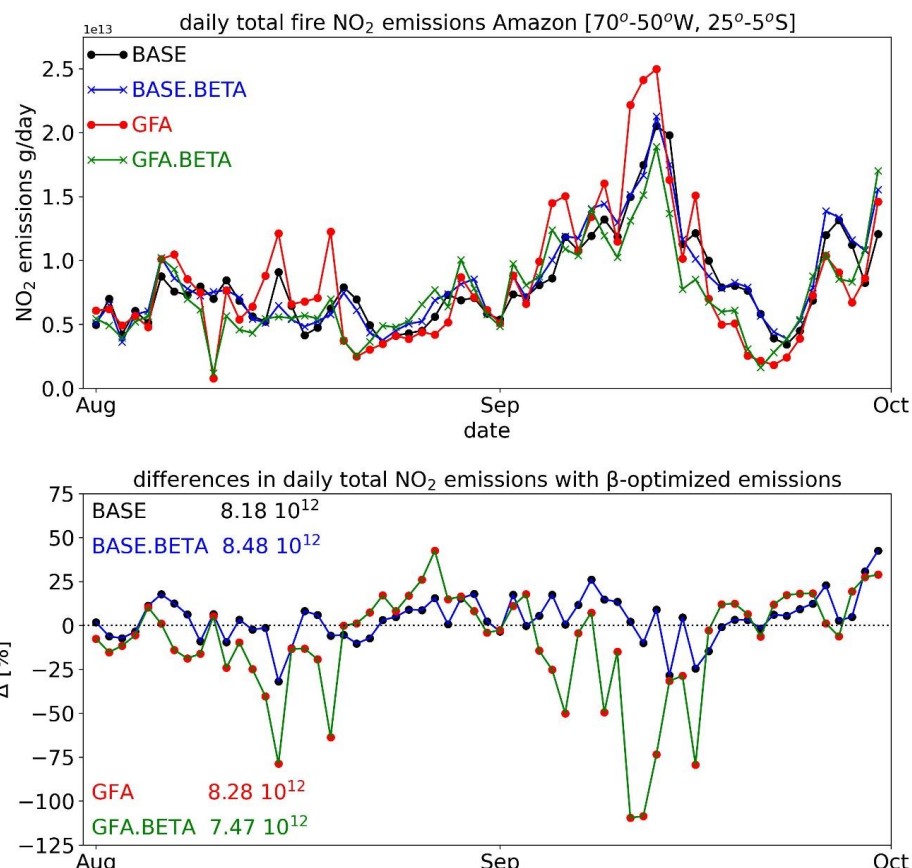

**Figure 9.** Daily total $NO_2$ emissions over the larger Amazon region for August and

September 2020. GFAS emissions in black (BASE), β-optimized GFAS emissions in blue

(BASE.BETA), GFA emissions in red (GFA), β-optimized GFA emissions in green

(GFA.BETA).

For Africa and the Siberia tundra regions – where only a comparison with β-optimized

emission is available for GFAS data – results are similar (Appendix Fig. A9). For the Siberia

steppe region absolute emissions (~0.11-0.12 Tg day$^{-1}$) are approximately an order of



magnitude smaller compared to those for the Siberia tundra region (0.75 Tg day$^{-1}$) and
approximately two orders of magnitude smaller than those for Africa and the Amazon (~8 Tg
day$^{-1}$). Comparing the various emission datasets with TROPOMI tropospheric NO$_2$
observations reveal a fair to strong spatial correlation ranging between 0.568 and 0.993
depending on emission database and regions (R$^2$, Pearson and Spearmon; see Appendix Table
A2), except for the Siberia steppe region. This reflects the limited number of fires in the Siberia
steppe region and a limited number of days for the Siberian steppe region where β-optimized
emissions differ from the GFAS emissions (Appendix Fig. A9) indicating that for most days
there are no fires and thus no NO$_x$ emission updates.

**4. Discussion**

Using bottom-up fire NO$_x$ and CO emission estimates in the IFS-COMPO model and then
comparing results with TROPOMI data revealed the existence of two significant biases in
bottom-up emission estimates. GFAS emissions were used as a "state of the art" global fire
emission database, the emission data developed in the S4F project in order to update bottom-
up fire emissions using remote sensing data not used in for example GFAS.
Overall, the results of the Amazon IFS NO$_x$ simulations and sensitivity tests can be
summarized as follows.
• IFS-COMPO simulations with GFAS emissions results in an overestimation of
tropospheric NO$_2$ columns over fire regions, especially for large fires, the so-called "large
tropospheric NO$_2$ column" bias
• IFS-COMPO simulations with GFAS systematically underestimate background
tropospheric NO$_2$ columns, possibly pointing to an underestimation of soil NO$_x$ emissions





- IFS-COMPO simulations GFA-S4F emissions worsen the "large tropospheric NO$_2$ column" bias resulting in a significantly larger structural IFS-COMPO NO$_2$ column bias
- capping the NO$_x$ emissions largely reduces the "large tropospheric NO$_2$ column" bias but worsens the statistics, in particular the regression of modeled vs. observed tropospheric NO$_2$ columns
- optimizing the fire NO$_2$ emissions based on TROPOMI NO$_2$ observations using the IFS-based β-method drastically improves the IFS-COMPO results, in particular for the GFA-S4F emissions, but does not completely close the gap between model and TROPOMI observations
- updating the soil emissions alone does not improve the IFS-COMPO simulations results
- combining the β-optimized fire emissions together with updated soil NO$_x$ emissions yielded the best results both in terms for correlations and regression coefficients

Note that the persistent "large-tropospheric-NO$_2$-column" bias regardless of using GFAS or GFA-S4F emissions implies its cause is not satellite-data based vegetation characteristics but possibly emission factors that are used to translate these vegetation characteristics to trace gas emission amounts. For example, NO$_x$ emissions associated with burning and combustion are strongly temperature dependent and highly non-linear. Only when combustion takes place at very high temperatures larger than 1500° Celsius can N$_2$ break down into atomic N that can recombine with O$_2$ to form NO and NO$_2$, the so-called Zeldovich mechanism. These are temperatures associated with the blue-flaming phase of fires. Given that laboratory measurements of fire NO$_x$ emission factors necessarily are restricted to small fires it is conceivable that those emission factors are not representative for large fires. Jin et al. (2021) recently showed that - using TROPOMI NO$_2$ data - fire NO$_x$ emission factors appear much more variable with a much larger dynamical range than currently assumed and used.



Secondly, based on the comparison of IFS-COMPO results and TROPOMI data, CO
emissions were consistently largely underestimated by GFAS for the four regions we explored
(biased low by 20 - 70%) even though the spatial correlation between observed and modeled
CO total columns was very good with correlations ($R^2$) exceeding 0.85 for all regions and cases.
Using the GFA-S4F CO emissions rather than the GFAS emissions for the Amazon improved
the spatial correlation while on average significantly decreasing the bias, possibly even
eradicating the bias depending on the method with which the data was evaluated. That does not
mean all issues were resolved as there was an approximately 25% standard deviation in the
differences of modeled and observed CO total columns indicating that locally discrepancies
between emissions and observations remain. Nevertheless, the results strongly suggest that the
(larger) CO emissions in GFA-S4F are more realistic than those from GFAS. Note that the
underestimation of background values of atmospheric CO is a common problem with many
atmospheric chemistry models (Gaubert et al., 2020), not just IFS-COMPO, and that this
underestimation likely has multiple causes (Inness et al., 2022).
Bottom-up fire CO emission estimates have for decades continued to be rather uncertain
for various reasons and despite significant amounts of research on the topic (Andreae, 2021).
Important culprits are the characterization of land cover types, fuel conditions as well as fire
dynamics and weather conditions. For many bottom-up fire-emission parameters there is
insufficient *in situ* data or empirical data and observations to constrain emissions. An important
source of uncertainty is the satellite observation-based characterization of land cover type.
While there are now many satellites observation Earth's surface and many more methods and
approaches to characterize the land cover type, considerable differences between land use and
land cover datasets remain (Liu et al., 2021; see further Khaldi et al. (2022) and references
therein). Another important source of uncertainty are satellite-based fuel loads and fuel





conditions. Observations to constrain these parameters are typically only available once every
10 days or worse as clouds can further limit satellite observations of these parameters.

Finally, although the first S4F results are very promising towards improving fire emissions,

the approach presented here is built on generic statistics: combining many fires and the effect
of many fires and reduce the analysis results to a few statistics. Although valuable, this
approach does not make optimal use of the rich information density of the satellite data. If
many uncertainties are related to fire specific properties and conditions then further refinement
and analysis of individual fires – as to some extent explored for $NO_2$ in Jin et al., (2021) -
would be a worthy approach. This, however, requires stepping away from gridded and averaged
data and change thinking towards fires as single and unique spatial structures. Each fire, each
structure, would be associated with specific characteristics: its fuel load, vegetation type(s),
fuel moisture, area, moisture, weather conditions, and emission plume characteristics. The
GFA-S4F data is a step in this direction as the emission data is provided per fire structure
(polygons) and each fire was associated with other fire characteristics from GFA-S4F data
based on Sentinel-2 and Sentinel-3 data. However, for TROPOMI data such an approach is still
lacking. Fire emission plumes would have to be identified first and then linked to a fire.
Automated detection of TROPOMI-based (fire) emission plumes has only started to be
developed in recent years (Kurchuba et al., 2021; Finch et al., 2022; Goudar et al., 2023; Schuit
et al., 2023), especially thanks to the recent advance of data-intensive artificial intelligence
analysis techniques, but has the potential to further advance satellite-data-based estimates of
fire emissions.
**5. Conclusions.**

The Sense4Fire project aims to increase the scientific understanding of fire dynamics and

their role in the carbon cycle by integrating observations from the Sentinels into new Earth





observation products. This paper presents a first analysis of TROPOMI satellite observations
of fire plumes affecting atmospheric composition, and the use of trace gas (CO, $NO_2$) from
TROPOMI together with IFS-COMPO model simulations and to evaluate and optimize
satellite-based fire CO and $NO_x$ emissions.

TROPOMI allows for observing single fire emission plumes ($NO_2$, CO, AAI) on a daily

basis with unprecedented accuracy and spatial resolution. Results show that CO and AAI
correlate very well, but not with $NO_2$, related to the much shorter lifetime of $NO_2$. Visually
there is also an excellent agreement with VIIRS RGB imagery.

The analysis of August-September 2020 daily TROPOMI data and IFS-COMPO model

results over the Amazon/Cerrado region reveal significant biases in bottom-up emission data
of CO and $NO_x$. For simulated $NO_2$ a significant positive bias for large-fire cases over the
Amazon/Cerrado region was identified attributed to the GFAS fire emissions, while CO
emissions were significantly underestimated. Note that total $NO_x$ emissions are dominated by
small fires with only a small contribution from the few large fires but that this large fire bias is
nevertheless of concern as it reflects a lack of understanding.

These biases could not be attributed to the IFS-COMPO model resolution or sub-grid plume

chemistry processes. When using fire emissions from the GFA-S4F system which incorporates
more advanced geo-information that tracks individual fires the evaluations against TROPOMI
CO total columns were significantly improved, but the $NO_2$ tropospheric column evaluations
worsened by showing an increased positive model bias. This suggests that not only there is a
considerable uncertainty in the dry-matter-burned estimates, but also in the emission factors
that define the ratio between CO and $NO_x$ emissions.



704 A scaling approach was adopted to constrain bottom-up fire $NO_x$ emissions with

705 TROPOMI $NO_2$ observations, which relies on the local sensitivity of tropospheric $NO_2$ column

706 changes with respect to $NO_x$ emission changes (the β-method). This brought the emission

707 variability much closer in line with those from GFAS, independent of which prior emission

708 estimate was used. Feeding any of the optimized fire $NO_x$ emissions back into the model indeed

709 led to a significant improvement and disappearance of the positive bias associated to large

710 emission sources, while the background model bias was unaffected. Combined with improved

711 soil $NO_x$ emissions results are on average further improved. This illustrates that emission types

712 of different origin can be optimized independently, and that both emission types need to be

713 optimized to match the model simulations with the observations.

714 Overall results presented here show that advanced use of geo-information from the suite of

715 ESA Sentinel satellites helps improve and constrain fire emissions, although not perse by

716 relying solely on satellite data-based bottom-up emissions – for instance a careful assessment

717 of emission factors is needed. On the other hand, the focus of this paper as well as the first

718 phase of the S4F project has been on average and cumulative statistics. Although those statistics

719 could be improved, that approach does not address many uncertainties and discrepancies at

720 local spatial scales and the level of individual fires. Also, in depth understanding of the biases

721 that were identified is still lacking and requires additional research. Fortunately, the suite of

722 ESA Sentinel satellites allows for much more detailed in-depth analysis of fires and the S4F

723 project will be extended to further explore its results and provide more detailed analyses of

724 fires and their contributing factors.

725 **Acknowledgements**




This research is funded by the ESA Sense4Fire project which is part of the Carbon Science
Cluster of ESA's Scientific Data Exploitation element of the Earth Observation Envelope
Programme (EOEP-5). The Sense4Fire project is funded by ESA under ESA Contract Number:
4000134840/21/I-NB. Sentinel-5 Precursor is a European Space Agency (ESA) mission on
behalf of the European Commission (EC). The TROPOMI payload is a joint development by
ESA and the Netherlands Space Office (NSO). The Sentinel-5 Precursor ground-segment
development has been funded by ESA and with national contributions from The Netherlands,
Germany, and Belgium. This work contains modified Copernicus Sentinel-5P TROPOMI data
(2018-2022), processed in the operational framework or locally at KNMI.

The authors thank - in alphabetical order - Alfred Awotwi (Cardiff University), Daniel
Kinalczyk, Christopher Marrs and Christine Wessollek (Technical University Dresden) as well
as ESA project officer Stephen Plummer for their contributions within the S4F project that led
to this paper.

*Author contributions.* A.d.L. wrote the paper and did the majority of data analysis and
interpretation. V.H. performed the IFS-COMPO model simulations while N.A. provided GFA
emission data. M.F., V.H and N.A. all reviewed the paper and contributed to the discussion
and interpretation of results.

*Competing interests.* The authors declare that they have no conflicts of interest.

*Data availability.* TROPOMI data used in this paper is available via the EU COPERNICUS
data space or Amazon Web Services. GFA emission data is available via de Global Fire Data
webportal. GFAS emission data is available via the EU COPERNICUS atmosphere data store.



IFS-COMPO model simulations are stored in the ECMWF archives and can be made available
on request.

https://dataspace.copernicus.eu/

https://registry.opendata.aws/sentinel5p/

https://www.globalfiredata.org/

https://ads.atmosphere.copernicus.eu/

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

**Appendix**

**Application of Sentinel-5p TROPOMI data for fire monitoring**

The launch of the Polar orbiting TROPOMI instrument on board of the Sentinel-5p (S5p)

satellite in October 2017 with at that time unprecedented spatial resolution, data accuracy and
precision has opened up a whole new range of possibilities for monitoring and studying fires.
Several research papers have been published in recent years exploring the use of TROPOMI
data for those particular applications.

Li et al. (2020) used TROPOMI CO data for a selection of 41 wildfires across the United

States for a 15-month period in 2018-2019 to assess geostationary GOES-R FRP-based fire
emissions. CO emissions from TROPOMI data were estimated using a CO column mass budget
approach. They found a very good agreement between emissions based on both methods for
this selection of US wildfires.

Van der Velde et al. (2021) presented a first analysis of daily TROPOMI $NO_2$/CO ratios

uncovering spatio-temporal differences "that point to distinct differences in biomass burning
behavior". Although they used daily (single fire) TROPOMI data they extracted data on



regional to continental scale regions to derive statistical relationships. Using chemistry-
transport model simulations and bottom-up GFEDv4 fire emissions they found the model
results to be broadly consistent with the TROPOMI observations.
Griffin et al. (2021) focused on a few selected North American fires in TROPOMI data to
estimate fire $NO_x$ emissions using plume model simulations while also comparing with *in situ*
field campaign data from the FIREX-AQ campaign. They found that there is a good agreement
between satellite observation and *in situ* data and that TROPOMI $NO_2$ data can be used to
determine single fire $NO_x$ emissions.
Jin et al. (2021) calculated vegetation-specific $NO_x$ emissions and FRP emission factors
based on a large selection of isolated single fire emission plumes using TROPOMI and
Gaussian plume modelling. They found significant differences between previously reported
and observed emission factors suggesting a much larger variability amongst different fires than
generally assumed.
Stockwell et al. (2022) used aircraft data to estimate fire emissions for five different fires
and compared them with geostationary observations of FRP and burned area as well as
emissions of carbon monoxide based on TROPOMI data. They found a strong correlation
between the emissions based on the *in-situ* data and the emissions based on the TROPOMI
data.
Griffin et al. (2023) use TROPOMI CO data to create a global database of single or local
fire burning CO emissions for the period 2019-2021 to avoid smoke and cloud obscuring
effects of FRP measured by satellite instruments like MODIS and VIIRS. In addition, they also
use TROPOMI CO data to derive emission factors ("emission coefficients"), *i.e.* the amount of
CO emission as a function of FRP. They find a large range of biome dependent emission factors





for different types of forests and conclude that simple biomes classifications for estimating fire
emissions are insufficient and that, if anything, further biomes distinction and refinement is
warranted. They also note that more information on the burning stage of a fire and temporal
fire development is crucial for improving fire emission estimates, which with the use of polar
orbiting satellite instruments is hampered by the once or twice per day overpass.
Wan et al. (2023) analyzed TROPOMI $NO_2$ and CO data for the massive Australian 2020
"New Year's bushfire event" with a focus on deriving emission ratios and emission factors for
different vegetation types. They note that TROPOMI data can help identify the relative
contributions of different flaming phases over larger regions.
All these studies highlight the potential of using TROPOMI data for assessing fire
emissions. They also all note that their studies are the first exploratory steps using TROPOMI
and that more research is needed and warranted while approaches could be expanded, extended
and refined.
The S4F project explores the suite of the Sentinel satellites using a novel synergetic
approach to derived global fire emissions based on the characterization of individual fires and
their behavior, eventually to better constrain total carbon emissions and emission factors. The
ESA Sentinels have a huge potential to observe and quantify fire dynamics in terms of pre-fire
surface conditions (vegetation cover and fuel moisture content), fire behavior (FRP, burned
area, fire size) and fire effects on the atmosphere (fire emissions of trace gases and aerosols).
However, this combined potential has not yet been exploited even though there is a clear need
for such an integrated synergetic approach.




**IFS COMPO**

The default tropospheric chemistry of IFS-COMPO as used here is based on CY48R1 as described in https://www.ecmwf.int/en/elibrary/81374-ifs-documentation-cy48r1-part-viii-atmospheric-composition. Organic chemistry for trace gases up to propane is modeled explicitly, while lumped tracers are used for specific types of functional groups to model the oxidation of higher volatile organic compounds (Huijnen et al., 2010; Williams et al., 2013). The updated isoprene oxidation parameterization is documented in Williams et al. (2022). Photolysis rates in the troposphere are computed using the modified band approach (MBA) (Williams et al., 2006, 2012). The tropospheric chemistry mechanism consists of 71 trace gases and 127 gas-phase reactions, 30 photolysis rates, 3 heterogeneous reactions and 2 aqueous phase reactions. It is solved based on Kinetic PreProcessor (KPP) routines, using the four stages and third-order Rosenbrock solver (Sandu and Sander, 2006).

The aerosol component in IFS-COMPO is described in Rémy et al. (2022) and is based on a bulk-bin aerosol scheme. It simulates mass mixing ratio of the tracers for sea salt, desert dust, organic matter (OM), black carbon (BC), sulfate, nitrate, ammonium, and secondary organic aerosol (SOA), and is coupled to the tropospheric chemistry scheme for the formation of secondary organic and inorganic aerosol. In all, the aerosol module consists of 16 tracers, which are subject to processes such as hygroscopic growth, ageing, sedimentation.



| IFS run ID | emissions | IFS version | IFS emission setup | IFS > 20 TROPOMI < 20 | | | IFS < 20 TROPOMI > 20 | | | IFS > 10 TROPOMI < 10 | | | IFS < 10 TROPOMI > 10 | | |
|---|---|---|---|---|---|---|---|---|---|---|---|---|---|---|---|
| | | | | N | mean ratio | median ratio | N | mean ratio | median ratio | N | mean ratio | median ratio | N | mean ratio | median ratio |
| BASE | GFASv1.4 | CY47R3.1 | | 60 | 8.65 | 7.66 | 2 | 2.94 | 2.94 | 260 | 5.46 | 4.28 | 45 | 2.74 | 2.56 |
| BASE.CAP0.1 | GFASv1.4 | CY47R3.1 | capped 0.1 mg m$^{-2}$ d$^{-1}$ | - | - | - | 3 | 7.50 | 7.50 | - | - | - | 57 | 5.65 | 5.47 |
| BASE.CAP0.3 | GFASv1.4 | CY47R3.1 | capped 0.3 mg m$^{-2}$ d$^{-1}$ | 2 | 3.30 | 3.30 | 3 | 4.14 | 4.21 | 19 | 3.05 | 2.48 | 56 | 3.68 | 3.51 |
| GFA | GFA-S4F | CY47R3.1 | | 497 | 14.24 | 11.10 | 2 | 3.16 | 3.16 | 1183 | 9.39 | 6.79 | 33 | 2.41 | 2.08 |
| GFA.CAP0.3 | GFA-S4F | CY47R3.1 | capped 0.3 mg m$^{-2}$ d$^{-1}$ | 9 | 8.02 | 5.71 | 3 | 4.36 | 4.61 | 87 | 4.88 | 3.94 | 55 | 3.36 | 3.13 |
| GFA.IFSCYCLE | GFA-S4F | CY48R1.0 | | 441 | 14.7 | 11.51 | 2 | 3.20 | 3.20 | 1034 | 9.66 | 7.07 | 37 | 2.40 | 2.22 |
| BASE.BETA | GFASv1.4 | CY48R1.0 | β-optimized | 27 | 12.15 | 11.96 | 2 | 1.58 | 1.58 | 67 | 7.23 | 4.58 | 37 | 2.35 | 1.97 |
| GFA.BETA | GFA-S4F | CY48R1.0 | β-optimized | 51 | 23.17 | 17.09 | 2 | 1.42 | 1.42 | 122 | 13.06 | 7.39 | 36 | 2.33 | 1.90 |
| GFA.SOIL | GFA-S4F | CY48R1.0 | updated soil NO$_x$ | 444 | 14.68 | 11.49 | 2 | 3.14 | 3.14 | 1072 | 9.54 | 7.00 | 37 | 2.34 | 2.18 |
| GFA.BETA.SOIL | GFA-S4F | CY48R1.0 | updated soil NO$_x$ β-optimized | 99 | 24.87 | 15.08 | 1 | 1.62 | 1.62 | 258 | 14.02 | 7.84 | 22 | 1.33 | 1.82 |

1237





**Table A1.** Overview of IFS-COMPO-COMPO simulations used in this paper for the larger

Amazon region: four-letter/number IFS-COMPO simulation ID, fire emission database,

other emission specifics and IFS-COMPOIFS-COMPOversion. Right columns indicate the

IFS-COMPO simulation comparison with TROPOMI data statistics of the ratio between

simulated and observed daily tropospheric $NO_2$ columns for certain data selections.

Indicated are the number of IFS-COMPO grids meeting the selection criteria (N) and the

mean and median ratios. Data selections: IFS-COMPO $> 20\times10^{15}$ molecules cm$^{-2}$ and

TROPOMI $< 20\times10^{15}$ molecules cm$^{-2}$; IFS-COMPO $< 20\times10^{15}$ molecules cm$^{-2}$ and

TROPOMI $> 20\times10^{15}$ molecules cm$^{-2}$ IFS-COMPO $> 10\times10^{15}$ molecules cm$^{-2}$ and

TROPOMI $< 10\times10^{15}$ molecules cm$^{-2}$; IFS-COMPO $< 10\times10^{15}$ molecules cm$^{-2}$ and

TROPOMI $> 10\times10^{15}$ molecules cm$^{-2}$





| | $R^2$ [PEARSON] | $R^2$ [SPEARMAN] | $R^2$ [PEARSON] | $R^2$ [SPEARMAN] |
|---|---|---|---|---|
| AMAZON | | | | |
| | NO$_2$ | | CO | |
| BASE vs BASE.BETA | 0.925 | 0.900 | | |
| GFA vs GFA.BETA | 0.774 | 0.757 | | |
| BASE vs GFA.BETA | 0.631 | 0.556 | | |
| BASE vs GFA | 0.709 | 0.586 | 0.680 | 0.539 |
| BASE.BETA vs GFA.BETA | 0.774 | 0.690 | | |
| GFA vs BASE.BETA | 0.689 | 0.568 | | |
| | | | | |
| | BASE vs BASE.BETA | | | |
| AFRICA | 0.978 | 0.976 | | |
| SIBERIA tundra | 0.908 | 0.845 | | |
| SIBERIA steppe | 0.249 | 0.309 | | |

**Table A2.** Spatial correlations of emissions databases used in this study for the four 20°×20°
degree regions (Table 2). Note that the "BASE" simulation uses GFAS emissions. The lower
three row contain the correlations for the non-Amazon regions for which only GFAS and β-
optimized GFAS is available.



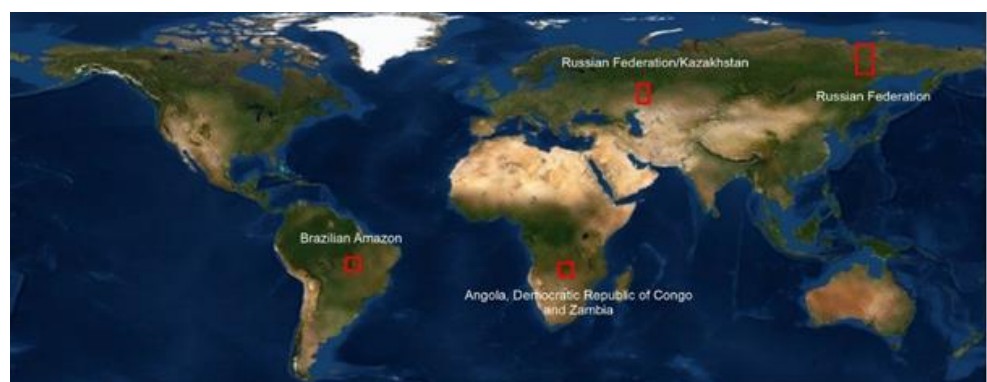


**Figure A1.** Location of the S4F test areas for the development of methods. Source: © Google
Maps 2024, Satellite Basemap, global view, https://www.google.com/maps/, 26-03-2024

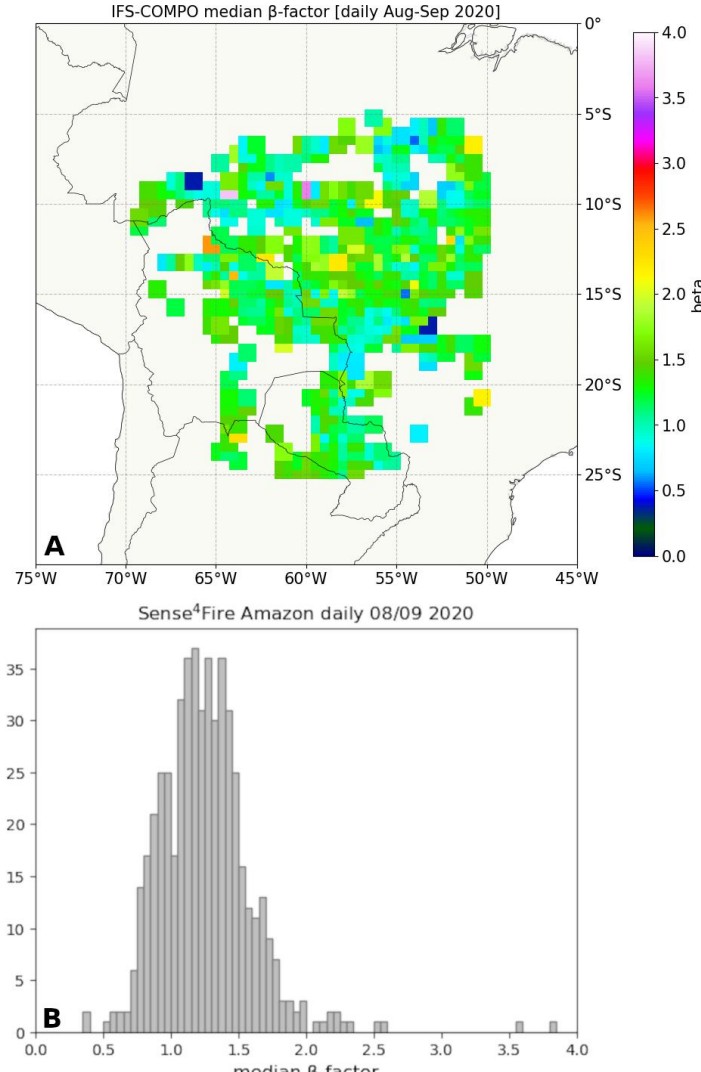

**Figure A2.** (A) Spatial distribution of IFS-COMPO values of the median β-factor over the Amazon region based on: [1] daily simulation data for August-September 2020 [2] for model grids with emissions larger than $1 \cdot 10^{-10}$ kg m$^{-2}$ s$^{-1}$ and [3] model grid NO$_2$ column values larger than $2 \times 10^{15}$ molecules cm$^{-2}$. (B) Histogram of data displayed in panel (A).

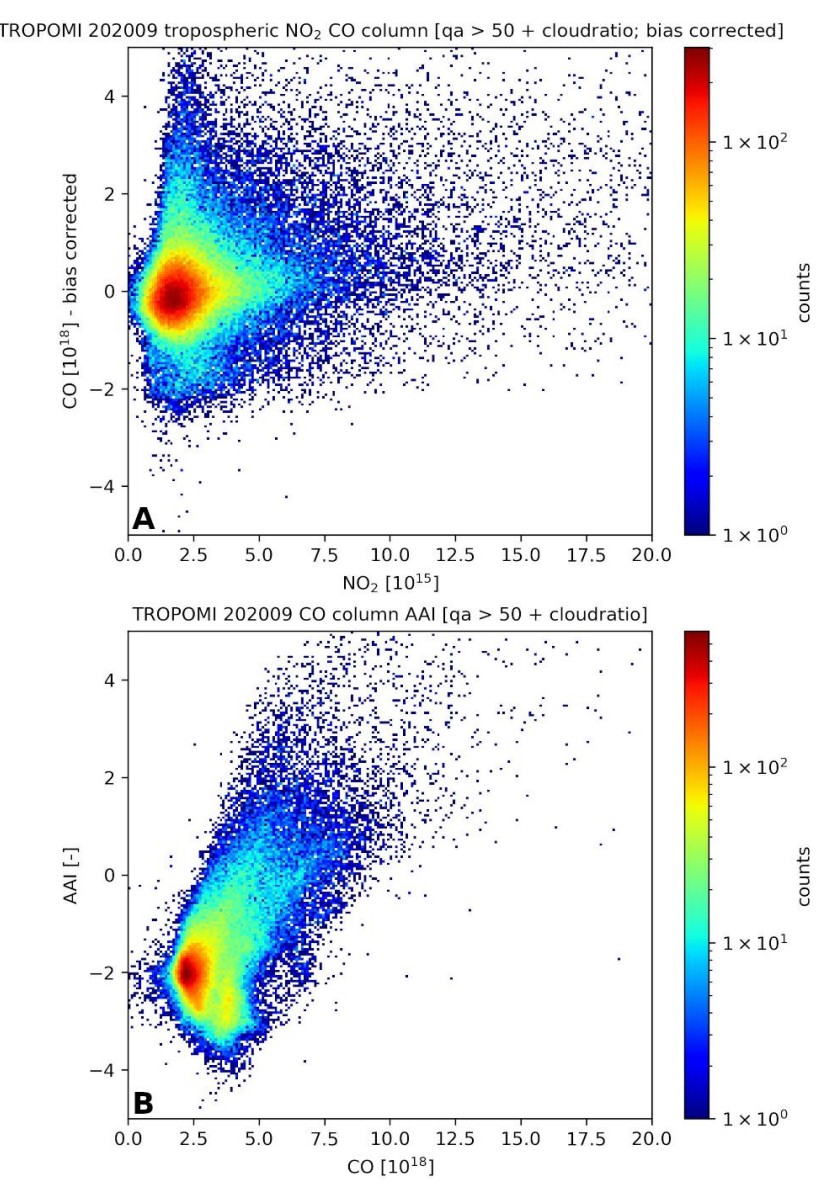

1263

**Figure A3.** (A) As Fig. 3, upper panel but without applying a daily median value bias correction for both the TROPOMI daily AAI and CO data. (B) as Fig. 3, middle panel,, upper panel but with applying a daily median value bias correction for TROPOMI CO data.



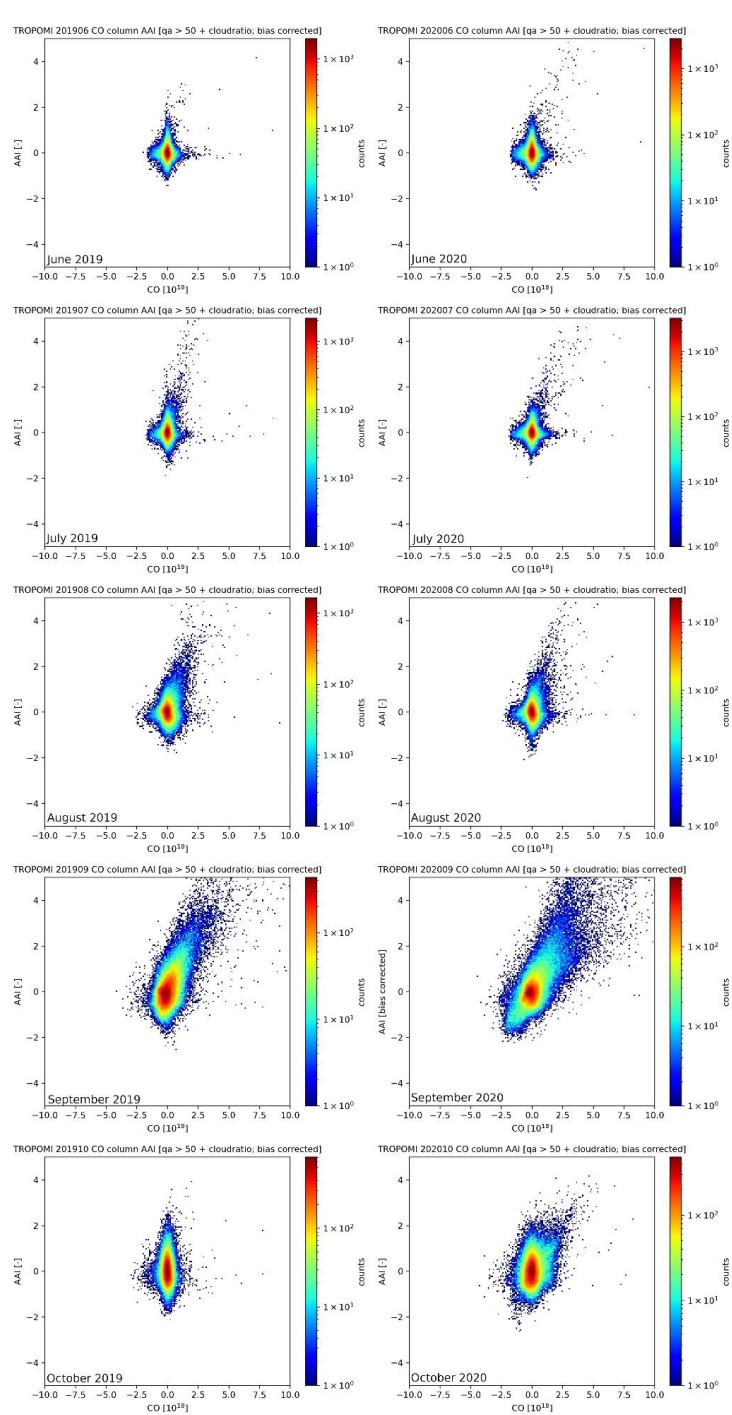



**Figure A4.** As Fig. 3, upper panel (2D histogram of TROPOMI NO$_2$ tropospheric columns and
CO total columns, the latter bias corrected using the daily median CO total column value) but
for all individual months between June and October and for both 2019 and 2020.







**Figure A5.** As Fig. 5 but for (A, B) the smaller Amazon region as displayed in Figs. 1 and 2; (C, D) as Fig. 5 but with GFA-S4F fire emissions; (E, F) as Fig. 5 but with GFA-S4F fire emissions and for IFS-COMPO version CY48R1; (G, H) As Fig. 5 but with GFA-S4F emissions over the Amazon region and with updated soil $NO_x$ emissions. Note that outside of the Amazon (25°S-EQ, 85°W-30°W) GFAS emission are used instead of the GFA-S4F emissions.








**Figure A6**. As Fig. 5 but for (A, B) GFAS fire emissions capped at $1 \cdot 10^{-10}$ kg m$^{-2}$ s$^{-1}$; (C, D)
GFAS fire emissions capped at $3 \cdot 10^{-10}$ kg m$^{-2}$ s$^{-1}$; (E, F) GFA-S4F fire emissions capped at
$3 \cdot 10^{-10}$ kg m$^{-2}$ s$^{-1}$; (G, H) β-optimized GFAS emissions.

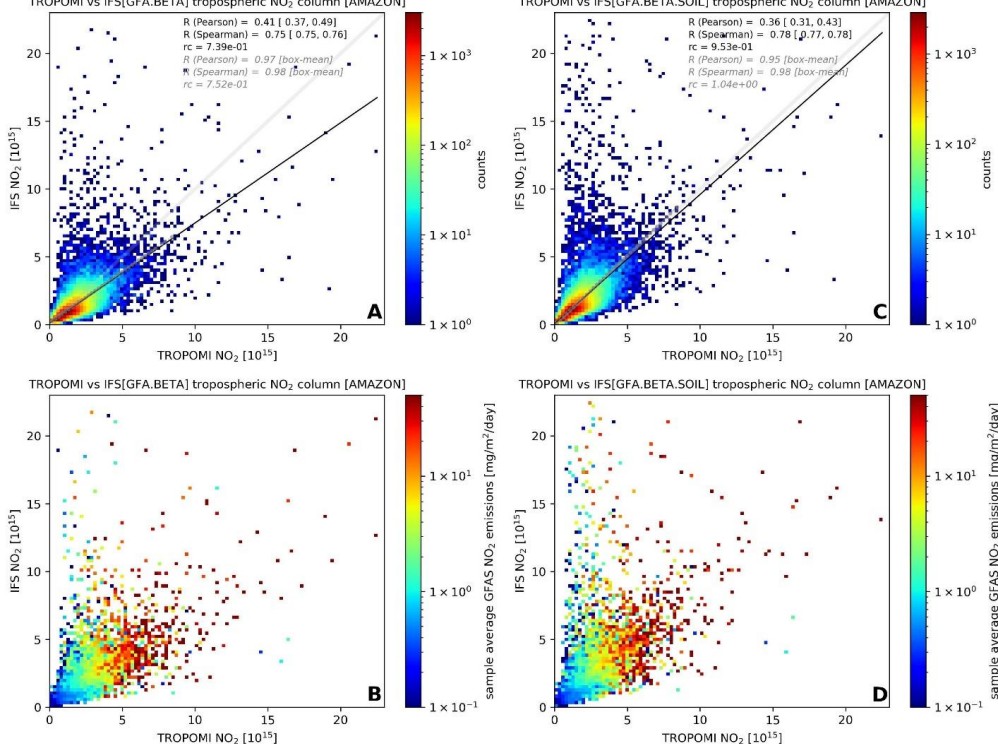


**Figure A7.** As Fig. 5 (A, B) but for (A, B) β-optimized GFA-S4F emissions and (C, D) β-
optimized GFA-S4F emissions and with updated soil NO$_x$ emissions








**Figure A8.** Comparison of IFS-COMPO and TROPOMI CO - similar to Figs. 6-7-8 but - for
four different regions based on GFAS emission (IFS-COMPO run BASE). For the Amazon
region also results from the IFS-COMPO simulation with GFA-S4F emissions are presented
(second panel; IFS-COMPO run GFA). Statistics are summarized in Table A2.





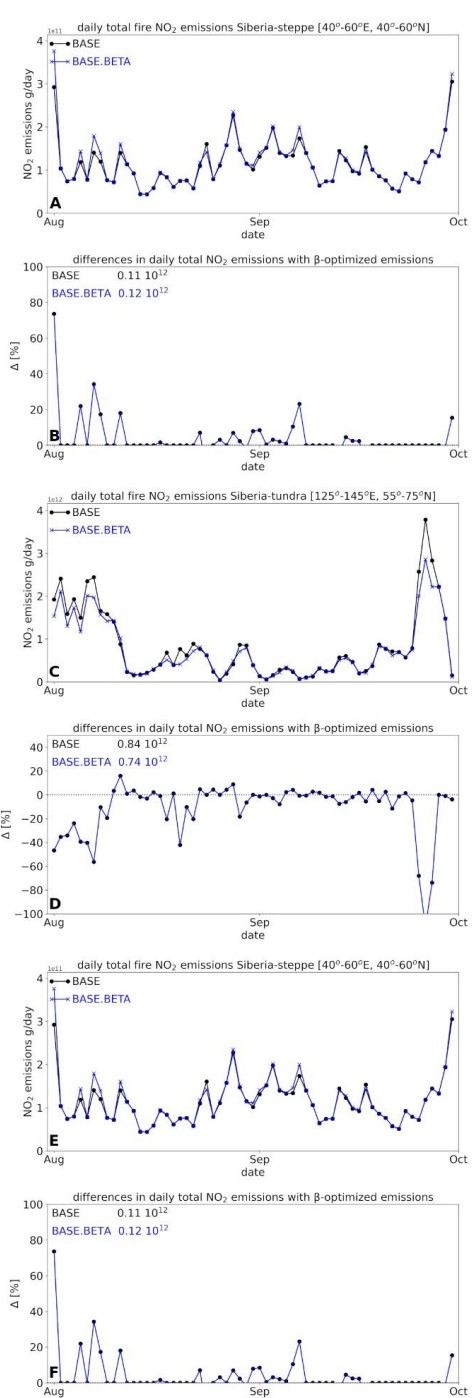







**Figure A9.** Absolute and relative differences in regional daily NO$_x$ emissions as in Fig. 9 but

for (A, B) the sub-equatorial Africa region, (C, D) the Siberia tundra region and (E, F) the

Siberia steppe region. See Appendix Fig. A1 for the location of these regions.