# Peer review of "Assessment of satellite observation-based wildfire emissions inventories using"

_EGUsphere, 2024_

## Referee Comment (RC3)

Review of "Assessment of satellite observation-1 based wildfire emissions inventories using TROPOMI data and IFS-COMPO model simulations," by A. de Laat et al.

In this paper, de Laat and coauthors compare model simulations of $NO_2$ and CO to satellite measurements in four regions that experience biomass burning. The goals of the paper appear to be (1) to validate the model simulation of biomass burning emissions and (2) to see if TROPOMI data can be used to constrain the model simulations, leading to a better match between model and measurements. For goal (2), the authors rely on what they call a "β-method" to adjust $NO_2$ emissions in their model. The method assumes that column amounts and emissions vary linearly with each other. The authors first scale the modeled $NO_2$ column amounts in a fire grid cell to match the observed TROPOMI column $NO_2$, and then the same scaling factor is then applied to emissions in that grid cell. The four regions investigated are an area in the Amazon Basin, a savannah in sub-Saharan Africa, a steppe region of Siberia, and a tundra region in Siberia. The paper focuses on a short time period, August-September 2020.

This paper could be of interest to the scientific community, as fire emissions in the Amazon Basin in particular are not well constrained. However there seemed to be little new science emerging from the paper, only (1) an acknowledgment of the difficulties inherent in model simulations of fire emissions and (2) an effort to show that the β-method improves the model match with observations. Model simulations for the Amazon also improved when soil $NO_x$ emissions were updated, but I could not find information on what went into these soil updates.

The paper requires major revision. The authors should strive to provide the reader with new and interesting scientific knowledge.

**Main criticisms.**

1. The authors use TROPOMI observations of $NO_2$ to adjust fire emissions so that the model results of column $NO_2$ better match the observations. There are two issues here.

First, the authors state that in regions characterized by thick smoke "no accurate tropospheric $NO_2$ column values . . . could be retrieved," as shown in Figure 2. Then Figures 6 and 7 show that in some cases the model predicts much greater $NO_2$ values than those observed by TROPOMI, and here the authors seem to blame the modeled emissions for these "overestimates." Is it possible that TROPOMI just cannot see the $NO_2$ in these regions? In that case, would it be misguided to adjust the model emissions using the β-method?

Second, the reader would like to know more exactly why the model emissions may be wrong. Why do these emissions need adjusting? What can developers of fire emissions learn from this exercise?

2. The paper is meant to showcase a new emissions inventory, Global Fire Atlas – Sense4Fire emissions (GFA-S4F). But there is little discussion about how GFA-S4F differs from the other emissions inventory tested, Global Fire Assimilation System (GFAS). What are the strengths and weaknesses of the two inventories? Is one inventory better than the other for certain ecoregions? This is information that would benefit the community.

3. The authors found a better match between model and observations when an updated soil NOx scheme was implemented in the model. The reader would like to know what is special about this new scheme, and whether it should be adapted globally.

4. Figures. Many of the figures could be relegated to the Supplement, as indicated in the minor comments. Captions are often incomplete, as noted below. Acronyms should be spelled out in caption. The extraneous and often mysterious text above the panels should be simplified so that the reader can quickly grasp what the figures show. Text in figures should be enlarged for readability. Date ranges should be given for all Figures. (It wasn't always clear if the Figure showed one day or an average over 2 months.)

5. Writing style. The introduction is well-written, but after that the writing becomes less clear, with 2-3 typos or lapses in English per page.

**Minor criticisms.**

1. The title refers to "wildfire emissions inventories" but most fires in the Amazon at least are not wild but deliberately set. In this region, fires are used for clearing land and for maintaining cropland.

2. The abstract and methods section describe using TROPOMI measurements of aerosol absorbing index (AAI), but the authors rely on AAI just to see if it correlates with CO. I recommend deleting mention of AAI in the abstract and perhaps also conclusions.

3. Lines 207-214. The description of the different-sized regions was confusing. It would be helpful to have a figure in the Supplement showing closeups of each region with the small and large domains clearly marked.

4. In the description of the atmospheric chemistry model, there is no mention of plume injection heights. How is this parameter handled in the model? If all fire emissions are injected into the surface model layer, what are the implications for the chemical lifetimes for $NO_2$ and CO? The authors should return to consideration of plume injection heights in the Discussion section.

5. Lines 324-325. The text states that "the β-method assumes that column amounts and emissions vary linearly which may not always be the case." It would be helpful to know what exactly may cause the column amounts and emissions to vary nonlinearly.

6. Figure 1. I think this Figure shows the smaller and larger domains of the study. If yes, this should be made clear in the caption. The reader is curious why the small region of Figure 1a is not centered in the larger region of Figure 1b. Is that because the prevailing winds carry smoke mostly westward?

7. Figure 2. It's very hard to see the open green circles. Are there also closed green circles? Caption should say what grey areas signify. Acronyms should be spelled out in caption. The text above each panel should be simplified so that the reader can quickly grasp what the figures show. Text in figure should be enlarged for readability.

8. Lines 367-368. Text says that chemical production and loss play a role in $NO_2$ concentrations, but there are no details in the paper about this chemistry.

9. Figure 3 should be in Supplement, as little new is learned.

10. Figure 4. What do all four lines mean in the bottom righthand panel? The abbreviation "rc" apparently stands for "regression coefficient," but I think a better term would be "slope," as the Pearson and Spearman $R$s are typically referred to as "regression coefficients." Instead of slope, however, the authors should consider reporting either normalized mean bias (NMB) or normalized mean error (NME). See Huang et al. (2021). NMB is a useful model performance indicator because it avoids over-inflating the observed range of values, especially at low concentrations. NME is similar to NMB, where the performance statistic is used as a normalization of the mean error.

Huang, L., et al. (2021). Recommendations on benchmarks for numerical air quality model applications in China – Part 1: $PM_{2.5}$ and chemical species, *Atmos. Chem. Phys., 21*, 2725–2743, https://doi.org/10.5194/acp-21-2725-202.

11. Figure 5. "Lower" in caption seems to be a typo.

12. Figure 6. The caption states that the two panels show the same distribution of points, but to my eye the distributions differ. See for example the two points at far right. In the caption, it is not clear what is meant by "bins" in the TROPOMI data. The grey line is the 1:1 line, right? If yes, that should be stated.

13. Figure 7. In my view, the color-coded plots showing the magnitude of fire emissions could all be relegated to the Supplement, as they do not add much.

14. Tables 1 and 2. I think these Tables were misplaced in the manuscript, with Table 2 shown before Table 1. Details of the sub-gridscale chemistry scheme for fire plumes should be discussed in the methods section. All acronyms should be defined, either in the captions or footnotes. See earlier comments regarding "regression coefficients." The methods section implies that simulations are performed in all four regions over both large and small domains. Why are results shown for only the larger domains in sub-Sahara Africa and the Siberia?

15. Line 523. What is meant by "dynamical range"?

16. Lines 522+. The sub-Saharan region is repeatedly referred to as "Africa." This seems to be a misnomer, given the wide variation in ecoregions across the continent.

17. Figure 18. See comments on earlier, similar figures.

18. Figure 19. Caption is really bare-bones. It's not clear how the differences were calculated.

19. Line 610. Text states that a certain configuration of the model yields "significantly larger structural IFS-COMPO $NO_2$ column bias." What is "structural" $NO_2$ bias?

21. Figure A1. Red squares are hard to see.

22. Figures in Supplement. See previous comments regarding Figures. Text in Figures is all too tiny.